# Image-to-Brain Signal Generation for Visual Prosthesis with CLIP Guided Multimodal Diffusion Models

**Ganxi Xu** [1]   **Zhao-Rong Lai** [*2]   **Yuting Tang** [*3 4]   **Yonghao Song** [*5]   **Guoxu Zhou** [6]   **Boyu wang** [7]   **Jian Zhu** [6]   **Jinyi Long** [1]

## Abstract

Visual prostheses hold great promise for restoring vision in blind individuals. While researchers have successfully utilized M/EEG signals to evoke visual perceptions during the brain decoding stage of visual prostheses, the complementary process of converting images into M/EEG signals in the brain encoding stage remains largely unexplored, hindering the formation of a complete functional pipeline. In this work, we present a novel image-to-brain signal framework that generates M/EEG from images by leveraging the diffusion transformer architecture enhanced with cross-attention mechanisms. Specifically, we employ a diffusion transformer (DiT) architecture based on denoising diffusion implicit models (DDIM) to achieve brain signal generation. To realize the goal of image-to-brain signal conversion, we use cross-attention mechanisms to align brain signal embeddings with CLIP image embeddings. Moreover, we leverage large language models (LLMs) to generate image captions, and concatenate the resulting CLIP text embeddings with CLIP image embeddings to form unified embeddings for cross-attention alignment, enabling our model to capture core semantic information. Furthermore, we introduce a learnable spatio-temporal position encoding that combines brain region embeddings with temporal embeddings to capture both spatial and temporal characteristics of brain signals. We evaluate the framework on two multimodal benchmark datasets (THINGS-EEG2 and THINGS-MEG) and demonstrate that it generates biologically plausible brain signals.

## 1. Introduction

Visual prostheses are advanced medical devices designed to restore partial vision for individuals with severe visual impairments or blindness, often caused by conditions such as retinitis pigmentosa (RP) and age-related macular degeneration (AMD) (Zrenner, 2013; Busskamp & Roska, 2011). These devices use an image sensor to capture external visual scenes and a processing framework to predict stimuli for an implanted electrode array (Humayun et al., 2012; Goetz & Palanker, 2016; Soltan et al., 2018) (we call this process brain encoding). The electrode array stimulates ganglion cells with the predicted stimuli, evoking visual perception (a pattern of localized light flashes, 'visual percept', or 'phosphene') in the retina (van der Grinten et al., 2024; Blom et al., 2010; Berry et al., 2017; Sahel et al., 2021; Granley et al., 2023) (this process is also referred to as brain decoding (Benchetrit et al., 2023)). The framework of visual prostheses is illustrated in Figure 1.

In the past few years, brain decoding has made significant progress (Lin et al., 2022; Scotti et al., 2023; Wang et al., 2024b; Li et al., 2024). Specifically, Mind-Reader (Lin et al., 2022), MindEye (Scotti et al., 2023), and Mind-Bridge (Wang et al., 2024b) utilize the high spatial resolution of functional magnetic resonance imaging (fMRI) to generate phosphenes. However, due to the high cost and low temporal resolution of fMRI limiting their applications in brain-computer interfaces (BCIs), Li *et al.* (Li et al., 2024) not only leverage the high temporal resolution of electroencephalography (EEG) signals to evoke visual percepts, but also demonstrate the versatility of their work on magnetoencephalography (MEG) signals. More importantly, these studies (Lin et al., 2022; Scotti et al., 2023; Wang et al., 2024b; Li et al., 2024) utilize multimodal datasets (Allen et al., 2022; Gifford et al., 2022) that include not only brain signals but also image data. Therefore, when training models, whether brain signals or image data are required, corre-

[1]College of Information Science and Technology, Jinan University, Guangzhou, China [2]Guangdong Key Laboratory of Data Security and Privacy Preserving, College of Cyber Security, Jinan University, Guangzhou, China [3]Department of Rehabilitation Medicine, The First Affiliated Hospital of Jinan University, Guangzhou, China [4]Center of Rehabilitation Medicine, Zhujiang Hospital, Southern Medical University, Guangzhou, 510280, China [5]Tsinghua University, Beijing, China [6]Guangdong University of Technology, Guangzhou, China [7]Western University, Ontario, Canada. Correspondence to: Jinyi Long <jinyil@jnu.edu.cn>.

*Proceedings of the 43rd International Conference on Machine Learning*, Seoul, South Korea. PMLR 306, 2026. Copyright 2026 by the author(s).

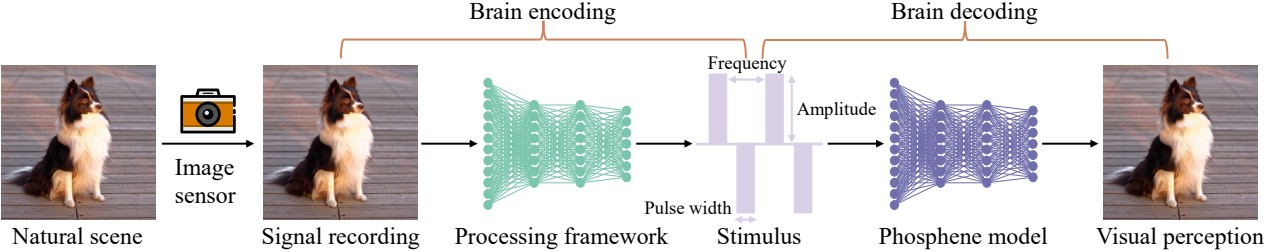

*Figure 1.* The framework of the visual prostheses. Visual prostheses utilize an image sensor to capture natural scenes. A processing framework takes the recorded signals as input and predicts the stimuli for the retinal prosthesis. A phosphene model receives stimulation from the implanted prosthesis and evokes visual perception (or 'phosphene'). The performance of the framework is evaluated by comparing the similarity between the original image and the visual perception.

sponding supervised signals can be provided to validate the model's output.

Compared to brain decoding, brain encoding has progressed slowly. For example, in his two papers (Granley et al., 2022; 2023), Granley uses the MNIST dataset (Deng, 2012) and the COCO dataset (Lin et al., 2014), both of which only contain image data. He takes the original images as supervised signals to find suitable predicted stimuli but does not use real stimuli as supervised signals to validate the accuracy of the predicted stimuli. Consequently, the limited biological resemblance of predicted stimuli confines the vision restoration effect of visual prostheses to rudimentary levels (Montazeri et al., 2019). To address this problem, Wang *et al.* (Wang et al.) use primary visual cortex (V1) responses as labels to find suitable predicted stimuli for better visual perception in the cortex. However, Wang *et al.* still do not use real stimuli as labels to evaluate the biological similarity of the predicted stimuli.

In addition to their practical value for visual prostheses, brain encoding models (also known as encoding models (Naselaris et al., 2011)) provide an explicit, quantitative framework for understanding how information is represented in brain signals (Naselaris et al., 2011). Brain encoding models enable researchers to test hypotheses about hierarchical visual processing, where deep neural network features have been shown to predict neural responses across the ventral visual stream, with early layers corresponding to V1 and deeper layers to inferotemporal cortex (Yamins & DiCarlo, 2016; Güçlü & Van Gerven, 2015; Raugel et al., 2025). Therefore, developing an accurate image-to-brain signal framework not only advances visual prosthesis technology but also provides computational tools for probing the neural mechanisms of visual perception (Kriegeskorte & Douglas, 2018).

To address the aforementioned issues, we propose an innovative image-to-brain framework that enables the conversion of images to M/EEG signals. First, we employ a diffusion transformer (DiT) (Peebles & Xie, 2023) to

achieve the generation of brain signals. To realize the conversion from images to brain signals, we use cross-attention mechanisms (Vaswani et al., 2017) to achieve the alignment between images and brain signals. Specifically, we use brain signal embeddings as the Query, and CLIP image embeddings as the Key and Value in the cross-attention block. Moreover, language descriptions of images can help the model learn core semantic information (Song et al., 2025). Therefore, we further use large language models (LLMs) to generate captions corresponding to the images, and then concatenate the CLIP text embeddings of these captions with the corresponding CLIP image embeddings to serve as the Key and Value in the cross-attention block. Meanwhile, we also note that brain signals contain both spatial and temporal information (Song et al., 2023). We employ a learnable spatio-temporal position encoding scheme. The position embedding consists of two components: (1) a region embedding that encodes which brain region each patch belongs to, and (2) a temporal embedding that encodes the temporal position of each patch. To validate our method's effectiveness, we conduct experiments on two multimodal datasets (THINGS-EEG2 (Gifford et al., 2022) and THINGS-MEG (Hebart et al., 2023)) containing both brain signals and image data. With these datasets, we can directly learn the mapping from images to brain signals using the ground truth brain responses as supervision signals.

Our main contributions are summarized as follows.

- We propose a novel image-to-brain framework that achieves the conversion of images to M/EEG signals using a diffusion transformer (DiT).

- We design a cross-attention mechanism that aligns images with brain signals by using brain signal embeddings as the Query, and the concatenation of CLIP image embeddings and CLIP text embeddings (derived from LLM-generated captions) as the Key and Value, enabling the model to capture both visual and semantic information.

- We introduce a learnable spatio-temporal position encoding scheme that explicitly models the spatial (brain region) and temporal characteristics of brain signals, allowing the model to better capture the inherent structure of M/EEG data.

- We conduct extensive experiments on two multimodal datasets (THINGS-EEG2 and THINGS-MEG), demonstrating the effectiveness of our proposed method in generating brain signals from images.

## 2. Related Works

### 2.1. Visual Prostheses

Visual prostheses are a promising treatment option for people living with incurable blindness (Ayton et al., 2020). The visual prostheses framework consists of two steps: The first step is brain encoding, which uses an image sensor to record natural scenes, then employs a processing framework to predict stimuli (Humayun et al., 2012; Goetz & Palanker, 2016; Soltan et al., 2018). The second step is brain decoding, which inputs the predicted stimuli into a phosphene model to evoke visual percepts (Berry et al., 2017; Sahel et al., 2021).

In recent years, brain decoding has made significant progress by leveraging the powerful generative capabilities of diffusion models (Lin et al., 2022; Scotti et al., 2023; Wang et al., 2024b; Li et al., 2024; Xu et al., 2023). In contrast, the development of brain encoding has progressed relatively slowly. Despite ongoing research efforts to improve the quality of predicted stimuli (Granley et al., 2022; 2023; Wang et al.), these studies fail to utilize real stimuli as supervised signals for evaluating the biological similarity of predicted stimuli, thereby limiting the vision restoration efficacy of visual prostheses to a low level (Montazeri et al., 2019).

To address the issues mentioned above, we use brain signals (M/EEG) from multimodal datasets (THINGS-EEG2, THINGS-MEG) as supervised signals to improve the biological similarity of predicted stimuli, thereby refining the image-to-brain framework.

### 2.2. Diffusion Models

Diffusion models (Ho et al., 2020; Dhariwal & Nichol, 2021; Du et al., 2023) have achieved remarkable success in generative modeling for their ability to generate high-quality images, in many cases surpassing Generative Adversarial Networks (GANs) (Goodfellow et al., 2020), which has previously represented the state-of-the-art. The seminal work on Denoising Diffusion Probabilistic Models (DDPMs) (Ho et al., 2020) introduces a U-Net architecture (Ronneberger et al., 2015) as the backbone for noise prediction. However,

Peebles *et al.* (Peebles & Xie, 2023) demonstrate that the U-Net inductive bias is not crucial to the performance of diffusion models and propose a Diffusion Transformers (DiT) architecture, which replaces the conventional U-Net with transformer architectures. Furthermore, a notable limitation of DDPMs is that they require simulating a Markov chain for many steps in order to produce a sample, resulting in slow generation speed. To address this inefficiency, Song *et al.* (Song et al., 2020) propose Denoising Diffusion Implicit Models (DDIMs), a more efficient class of iterative implicit probabilistic models that share the same training procedure as DDPMs while significantly accelerating the sampling process through a non-Markovian formulation.

Our goal is to convert images into brain signals. Therefore, we employ DiT based on DDIM to achieve the generation of brain signals. Subsequently, we utilize cross-attention to align brain signal embeddings with unified embeddings, which are formed by concatenating CLIP image embeddings and CLIP text embeddings, thereby achieving the objective of image-to-brain conversion.

### 2.3. EEG Signal Generation

Due to the difficulty in collecting EEG signals (Jiang et al., 2016) and the tremendous success of GANs in image generation (Goodfellow et al., 2016), researchers have turned their attention to using GANs to generate EEG signals for dataset augmentation (Hartmann et al., 2018; Luo et al., 2020). However, GANs are known to suffer from training instability (Arjovsky et al., 2017), which limits their effectiveness in generating reliable brain signals.

Given the limitations of GANs and the recent success of diffusion models in generating high-quality, diverse samples (Ho et al., 2020; Dhariwal & Nichol, 2021), we leverage DiT for brain signal generation. Since brain signals include not only EEG but also MEG, we develop a unified image-to-brain framework that can handle these two types of brain signals.

### 2.4. Large Language Model (LLMs)

Recently, Large Language Models (LLMs) have attracted considerable attention due to their exceptional performance across diverse natural language processing tasks. Moreover, several multimodal LLMs have exhibited strong cross-modal alignment between language and visual modalities, demonstrating impressive capabilities in image captioning, visual reasoning, and content generation, including the ability to describe and analyze various elements within images (Zhu et al., 2023). These results inspire us to use text descriptions to exploit image information. However, some LLMs require an enormous number of parameters, consuming substantial memory resources (Brown et al., 2020; Touvron et al., 2023). Fortunately, as LLMs con-

tinue to evolve through iterative updates, their performance has steadily improved while the number of parameters has been progressively reduced (Team et al., 2024; Bai et al., 2023a). Therefore, in this paper, we employ Qwen2-VL-2B-Instruct (Wang et al., 2024a; Bai et al., 2023b) to generate text descriptions for images.

## 3. Methodology

### 3.1. Problem Definition

Given an input image $\mathbf{x}_{\text{img}} \in \mathbb{R}^{C \times H \times W}$, our goal is to generate the corresponding brain signal $\mathbf{y}_{\text{brain}} \in \mathbb{R}^{N_c \times N_t}$, where $N_c$ represents the number of brain signal channels (63 for EEG, 271 for MEG) and $N_t$ denotes the temporal sampling points (250 for EEG, 200 for MEG). Formally, we aim to learn a mapping function $f : \mathcal{X}_{\text{img}} \to \mathcal{Y}_{\text{brain}}$ that can generate biologically plausible brain responses from visual inputs. The overall architecture of our proposed framework is illustrated in Figure 2.

### 3.2. DiT-based Brain Signal Generation with Cross-Modal Alignment

Our framework leverages a Diffusion Transformer (DiT) (Peebles & Xie, 2023) architecture based on Denoising Diffusion Implicit Models (DDIM) (Song et al., 2020) to achieve brain signal generation. To realize the conversion from images to brain signals, we employ cross-attention mechanisms to align brain signal embeddings with unified visual-semantic embeddings.

#### 3.2.1. DIFFUSION TRANSFORMER ARCHITECTURE

We adopt the Diffusion Transformer (DiT) architecture (Peebles & Xie, 2023), which leverages the transformer-based design for diffusion models. The DiT architecture offers better scalability and has demonstrated superior performance in generative tasks. Our DiT model $\boldsymbol{\epsilon}_\theta$ takes as input the noisy brain signal $\mathbf{y}_t$, the diffusion timestep $t$, and the conditioning information, and predicts the noise to be removed:

$$\hat{\boldsymbol{\epsilon}} = \boldsymbol{\epsilon}_\theta(\mathbf{y}_t, t, \mathbf{c}),$$

where $\mathbf{c}$ represents the conditioning embeddings derived from the input image.

We employ DDIM (Song et al., 2020) as our diffusion framework, which provides faster sampling compared to DDPM (Ho et al., 2020) while maintaining generation quality. The forward diffusion process gradually adds Gaussian noise to the brain signal:

$$q(\mathbf{y}_t|\mathbf{y}_0) = \mathcal{N}(\mathbf{y}_t; \sqrt{\bar{\alpha}_t}\mathbf{y}_0, (1 - \bar{\alpha}_t)\mathbf{I}),$$

where $\bar{\alpha}_t = \prod_{s=1}^{t}(1 - \beta_s)$ and $\{\beta_t\}_{t=1}^{T}$ is a variance schedule. The DDIM sampling process enables deterministic

generation:

$$\mathbf{y}_{t-1} = \sqrt{\bar{\alpha}_{t-1}} \left( \frac{\mathbf{y}_t - \sqrt{1 - \bar{\alpha}_t}\boldsymbol{\epsilon}_\theta(\mathbf{y}_t, t, \mathbf{c})}{\sqrt{\bar{\alpha}_t}} \right)$$
$$+ \sqrt{1 - \bar{\alpha}_{t-1}}\boldsymbol{\epsilon}_\theta(\mathbf{y}_t, t, \mathbf{c}).$$

#### 3.2.2. CROSS-ATTENTION FOR CROSS-MODAL ALIGNMENT

To achieve alignment between brain signals and visual information, we employ cross-attention mechanisms (Vaswani et al., 2017) within the DiT architecture. In each DiT block, we incorporate a cross-attention layer where the brain signal embeddings serve as the Query, and the unified embeddings (detailed in Section 3.2.3) serve as the Key and Value:

$$\mathbf{Q} = \mathbf{H}_{\text{brain}}\mathbf{W}_Q,$$
$$\mathbf{K} = \mathbf{c}_{\text{unified}}\mathbf{W}_K,$$
$$\mathbf{V} = \mathbf{c}_{\text{unified}}\mathbf{W}_V,$$
$$\text{CrossAttention}(\mathbf{Q}, \mathbf{K}, \mathbf{V}) = \text{softmax}\left( \frac{\mathbf{Q}\mathbf{K}^T}{\sqrt{d_k}} \right) \mathbf{V},$$

where $\mathbf{H}_{\text{brain}} \in \mathbb{R}^{N \times D}$ represents the brain signal patch embeddings, $\mathbf{c}_{\text{unified}} \in \mathbb{R}^{(N_{\text{img}} + N_{\text{txt}}) \times D}$ represents the unified visual-semantic embeddings (the meanings of $N_{\text{img}}$ and $N_{\text{txt}}$ are detailed in Section 3.2.3), $\mathbf{W}_Q, \mathbf{W}_K, \mathbf{W}_V \in \mathbb{R}^{D \times D}$ are learnable projection matrices, and $d_k$ is the dimension of the key vectors.

#### 3.2.3. UNIFIED VISUAL-SEMANTIC EMBEDDINGS

To capture both visual and semantic information from input images, we construct unified embeddings by combining CLIP image embeddings with CLIP text embeddings derived from LLM-generated captions.

**CLIP Image Embeddings:** We employ the Vision Transformer variant of CLIP (ViT-L/14) (Radford et al., 2021) as our visual encoder $\mathcal{E}_{\text{CLIP}}^{\text{img}}$ to extract patch-level visual representations from input images:

$$\mathbf{h}_{\text{img}} = \mathcal{E}_{\text{CLIP}}^{\text{img}}(\mathbf{x}_{\text{img}}) \in \mathbb{R}^{N_{\text{img}} \times D},$$

where $N_{\text{img}} = 257$ represents the number of image tokens (1 CLS token + 256 patch tokens) and $D = 768$ is the embedding dimension.

**LLM-Generated Captions and CLIP Text Embeddings:** Language descriptions of images help the model learn core semantic information (Song et al., 2025). Therefore, we leverage large language models (LLMs) to generate descriptive captions for each image. Specifically, we employ Qwen2-VL-2B-Instruct (Wang et al., 2024a) to generate detailed image descriptions:

$$\mathbf{s}_{\text{caption}} = \text{LLM}(\mathbf{x}_{\text{img}}),$$

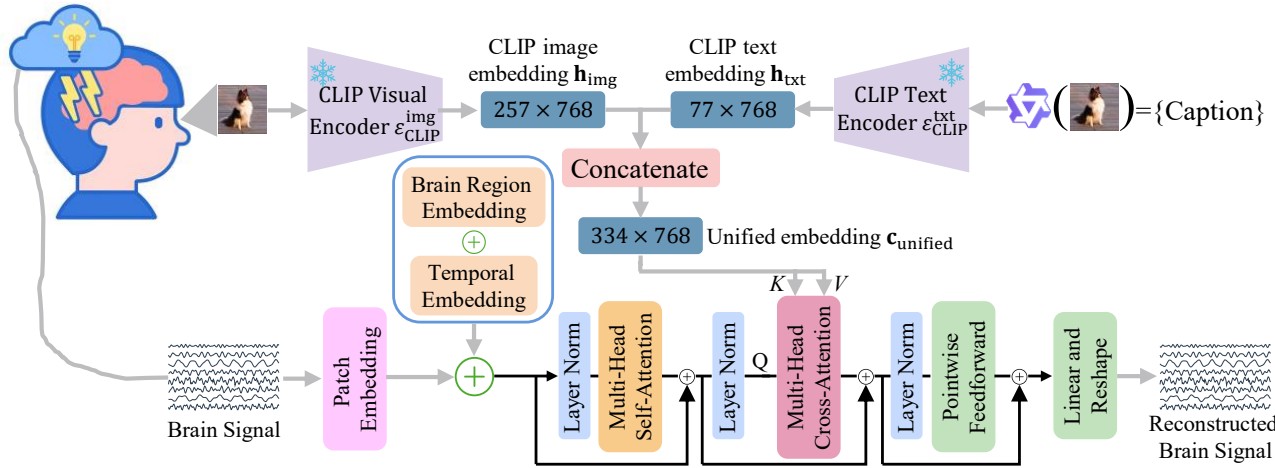

*Figure 2.* Overall architecture of our image-to-brain framework. The framework consists of: (1) a CLIP visual encoder (ViT-L/14) that extracts CLIP image embeddings, (2) a large language model (Qwen2-VL-2B-Instruct) that generates image captions which are then encoded by CLIP text encoder to obtain CLIP text embeddings, (3) the concatenation of CLIP image and text embeddings along the token dimension to form unified visual-semantic embeddings, and (4) a Diffusion Transformer (DiT) based on DDIM that generates brain signals through cross-attention mechanisms, where brain signal patch embeddings serve as Query and unified embeddings serve as Key and Value. The model incorporates learnable spatio-temporal position embeddings that combine brain region embeddings and temporal embeddings to capture the spatio-temporal characteristics of brain signals (detailed encoding methods are provided in Appendix B).

where $\mathbf{s}_{\text{caption}}$ is the generated text caption. We then encode these captions using the CLIP text encoder $\mathcal{E}_{\text{CLIP}}^{\text{txt}}$:

$$\mathbf{h}_{\text{txt}} = \mathcal{E}_{\text{CLIP}}^{\text{txt}}(\mathbf{s}_{\text{caption}}) \in \mathbb{R}^{N_{\text{txt}} \times D},$$

where $N_{\text{txt}} = 77$ represents the maximum number of text tokens.

**Unified Embeddings:** Finally, we concatenate the CLIP image embeddings and CLIP text embeddings along the token dimension to form unified embeddings:

$$\mathbf{c}_{\text{unified}} = \text{Concat}(\mathbf{h}_{\text{img}}, \mathbf{h}_{\text{txt}}) \in \mathbb{R}^{(N_{\text{img}}+N_{\text{txt}}) \times D},$$

which serve as the Key and Value in the cross-attention mechanism. This unified representation enables the model to leverage both visual information and the core semantic information of images for generating biologically plausible brain signals.

### 3.2.4. LEARNABLE SPATIO-TEMPORAL POSITION EMBEDDINGS

Visual processing follows a bottom-up hierarchy: visual stimuli are processed sequentially by V1, V2, V4 in the occipital cortex, and the inferotemporal cortex in the temporal cortex along the ventral stream for object recognition (Song et al., 2023). This hierarchical processing implies that brain signals inherently contain both spatial information (responses distributed across different brain regions from occipital to temporal cortex) and temporal information (sequential activation patterns over time). To capture

these spatio-temporal characteristics, we introduce a learnable position embedding scheme that combines brain region embeddings with temporal embeddings.

**Brain Region Embeddings:** We partition the brain signal channels into distinct brain regions and learn a region embedding for each brain region:

$$\mathbf{e}_{\text{region}} = \text{RegionEmbed}(r) \in \mathbb{R}^{D},$$

where $r \in \{1, 2, \dots, R\}$ denotes the region index and $R$ is the total number of regions.

**Temporal Embeddings:** To encode the temporal position of each patch along the time axis, we learn temporal embeddings:

$$\mathbf{e}_{\text{temporal}} = \text{TemporalEmbed}(\tau) \in \mathbb{R}^{D},$$

where $\tau \in \{1, 2, \dots, T_p\}$ denotes the temporal patch index and $T_p$ is the number of temporal patches.

**Combined Position Embeddings:** The final position embedding for each patch is obtained by summing the brain region embedding and temporal embedding:

$$\mathbf{e}_{\text{pos}} = \mathbf{e}_{\text{region}} + \mathbf{e}_{\text{temporal}}.$$

This additive combination allows the model to disentangle spatial and temporal information while maintaining a compact representation. The position embeddings are added to the patch embeddings before feeding into the DiT blocks, enabling the model to be aware of both the brain region and temporal context of each patch during the denoising process.

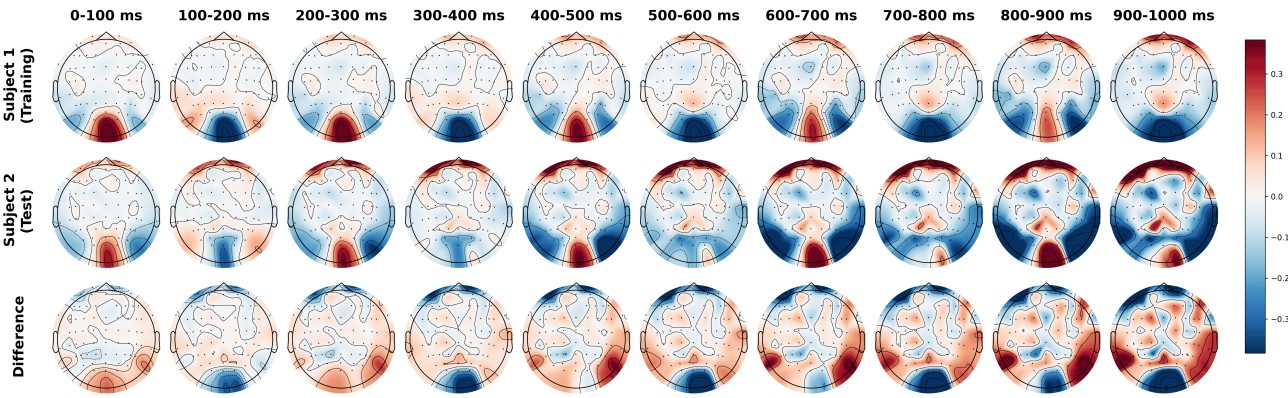

*Figure 3.* Cross-subject topography comparison for THINGS-EEG2. The figure shows EEG topographies comparing Subject 1 (training) and Subject 2 (test). The third row shows the difference between training and test subjects, highlighting inter-individual variability in brain signals.

### 3.3. Training Objective

During training, our model learns to predict the noise $\epsilon$ that was added to the clean brain signal. The training objective follows the standard diffusion model formulation:

$$\mathcal{L} = \mathbb{E}_{t,\mathbf{y}_0,\epsilon} \left[ \| \epsilon - \epsilon_\theta(\mathbf{y}_t, t, \mathbf{c}_{\text{unified}}) \|_2^2 \right],$$

where $\epsilon \sim \mathcal{N}(\mathbf{0}, \mathbf{I})$ is the sampled noise, $\mathbf{y}_t$ is the noisy brain signal at timestep $t$, and $\mathbf{c}_{\text{unified}}$ is the unified visual-semantic embedding.

### 3.4. Inference

During inference, given an input image $\mathbf{x}_{\text{img}}$, we first extract the unified embeddings by: (1) obtaining CLIP image embeddings $\mathbf{h}_{\text{img}} = \mathcal{E}_{\text{CLIP}}^{\text{img}}(\mathbf{x}_{\text{img}})$, (2) generating captions using the LLM and encoding them to obtain $\mathbf{h}_{\text{txt}} = \mathcal{E}_{\text{CLIP}}^{\text{txt}}(\text{LLM}(\mathbf{x}_{\text{img}}))$, (3) concatenating to form $\mathbf{c}_{\text{unified}} = \text{Concat}(\mathbf{h}_{\text{img}}, \mathbf{h}_{\text{txt}})$. We then start from pure Gaussian noise $\mathbf{y}_T \sim \mathcal{N}(\mathbf{0}, \mathbf{I})$ and iteratively denoise using the DDIM sampling procedure with the unified embeddings as conditioning. The final output $\mathbf{y}_0$ represents the generated brain signal corresponding to the input image.

## 4. Experiment

### 4.1. Performance Evaluation

We evaluate the quality of generated brain signals using four metrics: Mean Squared Error (MSE), Pearson Correlation Coefficient, Cosine Similarity, Synchronization Likelihood (SL) (Stam & Van Dijk, 2002).

We evaluate under two settings: (1) *Within-subject*, where the model is trained and tested on the same subject's data; (2) *Cross-subject*, where the model is trained on one subject and tested on different subjects, evaluating cross-subject generalization. For THINGS-EEG2 (Table 1), within-subject

*Table 1.* Quantitative evaluation on THINGS-EEG2

| Method | Evaluation Metrics | | | |
|---|---|---|---|---|
| | MSE ↓ | Pearson ↑ | Cosine ↑ | SL ↑ |
| Güçlü *et al.* (Güçlü & Van Gerven, 2015) | 0.211 | 0.178 | 0.181 | 0.458 |
| Yamins *et al.* (Yamins et al., 2014) | 0.196 | 0.203 | 0.209 | 0.476 |
| UNet-based Diffusion | 0.217 | 0.188 | 0.185 | 0.471 |
| MindSimulator (Bao et al., 2025) | 0.160 | 0.349 | 0.345 | 0.543 |
| SynBrain (Mai et al., 2025) | 0.156 | 0.366 | 0.347 | 0.577 |
| Our work | **0.109** | **0.425** | **0.402** | **0.614** |
| LOSO (Within-subject) | 0.200 | 0.293 | 0.307 | 0.535 |
| Our work (Cross-subject) | 0.174 | 0.157 | 0.165 | 0.522 |
| LOSO (Cross-subject) | 0.285 | 0.209 | 0.220 | 0.493 |

*Table 2.* Quantitative evaluation on THINGS-MEG

| Method | Evaluation Metrics | | | |
|---|---|---|---|---|
| | MSE ↓ | Pearson ↑ | Cosine ↑ | SL ↑ |
| Güçlü *et al.* (Güçlü & Van Gerven, 2015) | 0.831 | 0.243 | 0.254 | 0.405 |
| Yamins *et al.* (Yamins et al., 2014) | 0.796 | 0.269 | 0.274 | 0.423 |
| UNet-based Diffusion | 0.763 | 0.122 | 0.127 | 0.389 |
| MindSimulator (Bao et al., 2025) | 0.755 | 0.281 | 0.274 | 0.472 |
| SynBrain (Mai et al., 2025) | 0.727 | 0.302 | 0.326 | 0.514 |
| Our work | **0.663** | **0.379** | **0.382** | **0.597** |
| LOSO (Within-subject) | 1.015 | 0.237 | 0.215 | 0.437 |
| Our work (Cross-subject) | 0.768 | 0.082 | 0.089 | 0.461 |
| LOSO (Cross-subject) | 1.237 | 0.164 | 0.157 | 0.411 |

results are averaged over ten subjects, while cross-subject results are obtained by training on Subject 1 and testing on Subjects 2–10. For THINGS-MEG (Table 2), within-subject results are averaged over four participants, while cross-subject results are obtained by training on Participant 1 and testing on Participants 2–4.

**Within-subject Performance:** As shown in Tables 1 and 2, our method consistently outperforms all baseline methods across all evaluation metrics on both datasets. Compared to traditional encoding models (Güçlü & Van Gerven, 2015; Yamins et al., 2014) that use Convolutional Neural Network (CNN) features to predict brain signals, our approach achieves substantially better performance. Furthermore, our method also outperforms the recent genera-

tive approaches, including the UNet-based Diffusion baseline, MindSimulator (Bao et al., 2025), and SynBrain (Mai et al., 2025), demonstrating consistent improvements across all metrics. These results demonstrate that our DiT-based framework with cross-attention mechanisms and unified visual-semantic embeddings effectively captures the mapping from visual stimuli to brain signals.

**Cross-subject Generalization:** The cross-subject results reveal the challenge of inter-individual variability in brain signals. To visualize this variability, Figure 3 compares the topographies of Subject 1's training data with Subject 2's test data for EEG. The difference topographies reveal that these inter-subject variations are present across all time windows, explaining the performance degradation in cross-subject settings. The MEG cross-subject topography comparison is provided in Appendix H.

**Leave-One-Subject-Out (LOSO) Analysis:** We further adopt a leave-one-subject-out protocol (details in Appendix E). On both datasets, the LOSO cross-subject results consistently outperform the per-subject cross-subject results (e.g., Pearson: $0.157 \rightarrow 0.209$ on THINGS-EEG2, $0.082 \rightarrow 0.164$ on THINGS-MEG), confirming that incorporating multi-subject data effectively enhances cross-subject generalization. However, the LOSO within-subject results are lower than the per-subject within-subject results on both datasets, suggesting the need to explore models that are more robust to larger data volumes.

### 4.2. Ablation Study

*Table 3.* Ablation study on THINGS-EEG2

| Method | Evaluation Metrics | | | |
|---|---|---|---|---|
| | MSE ↓ | Pearson ↑ | Cosine ↑ | SL ↑ |
| w/o CLIP text embeddings | 0.117 | 0.405 | 0.388 | 0.608 |
| w/o Brain region embeddings | 0.136 | 0.383 | 0.365 | 0.599 |
| w/o Temporal embeddings | 0.120 | 0.395 | 0.384 | 0.587 |
| Our work | **0.109** | **0.425** | **0.402** | **0.614** |

*Table 4.* Ablation study on THINGS-MEG

| Method | Evaluation Metrics | | | |
|---|---|---|---|---|
| | MSE ↓ | Pearson ↑ | Cosine ↑ | SL ↑ |
| w/o CLIP text embeddings | 0.677 | 0.361 | 0.370 | 0.582 |
| w/o Brain region embeddings | 0.698 | 0.337 | 0.343 | 0.561 |
| w/o Temporal embeddings | 0.683 | 0.349 | 0.356 | 0.548 |
| Our work | **0.663** | **0.379** | **0.382** | **0.597** |

In this subsection, we conduct ablation studies to assess the effectiveness of the three key components in our framework: CLIP text embeddings, brain region embeddings, and temporal embeddings. First, we present the results obtained using the full model (Our work). Then, based on the full model, we remove the CLIP text embeddings (w/o CLIP text embeddings), so that the unified embeddings only con-

tain CLIP image embeddings. For brain region embeddings and temporal embeddings, we remove them from the position encoding separately (w/o Brain region embeddings and w/o Temporal embeddings, respectively). Tables 3 and 4 demonstrate that the removal of any component results in performance decline across all metrics, which shows the efficacy of our proposed components.

### 4.3. LLM-Generated Caption Quality Analysis

*Table 5.* Comparison of different LLMs for caption generation on THINGS-EEG2

| LLM | Evaluation Metrics | | | |
|---|---|---|---|---|
| | MSE ↓ | Pearson ↑ | Cosine ↑ | SL ↑ |
| VisualGLM (Du et al., 2022) | 0.147 | 0.370 | 0.350 | 0.578 |
| MiniGPT-4 (Zhu et al., 2023) | 0.132 | 0.386 | 0.369 | 0.596 |
| Qwen2-VL-2B-Instruct (Ours) | **0.109** | **0.425** | **0.402** | **0.614** |

*Table 6.* Comparison of different LLMs for caption generation on THINGS-MEG

| LLM | Evaluation Metrics | | | |
|---|---|---|---|---|
| | MSE ↓ | Pearson ↑ | Cosine ↑ | SL ↑ |
| VisualGLM (Du et al., 2022) | 0.705 | 0.308 | 0.317 | 0.522 |
| MiniGPT-4 (Zhu et al., 2023) | 0.689 | 0.341 | 0.350 | 0.574 |
| Qwen2-VL-2B-Instruct (Ours) | **0.663** | **0.379** | **0.382** | **0.597** |

To investigate the impact of different LLMs on brain signal generation, we compare Qwen2-VL-2B-Instruct (2B parameters) with two other multimodal LLMs: VisualGLM (Du et al., 2022) (6B parameters) and MiniGPT-4 (Zhu et al., 2023) (7B parameters). Qwen2-VL-2B-Instruct consumes fewer computational resources while still outperforming both baselines on both datasets, as shown in Tables 5 and 6.

To further evaluate the quality of LLM-generated captions, we compute the CLIP Score (Hessel et al., 2021) between each image and its corresponding caption, which measures the semantic alignment between visual and textual content (higher scores indicate better alignment). We evaluate all images from both training and test sets: 16,740 images for THINGS-EEG2 and 22,448 images for THINGS-MEG. Figure 4 shows an example of a caption generated by Qwen2-VL-2B-Instruct, demonstrating that the model produces a descriptive and semantically accurate caption for the image content.

As shown in Figure 5, the generated captions demonstrate high semantic relevance to the original images. The CLIP Score measures semantic alignment between images and captions, where higher scores indicate better alignment. For THINGS-EEG2, the mean CLIP Score is 0.6419, with 98.7% of captions achieving scores in the 0.6–0.8 range. Similarly, for THINGS-MEG Participant 1, the mean CLIP Score is 0.6421, with 98.7% of captions falling within the 0.6–0.8 range. Only 1.3% of captions have scores in the

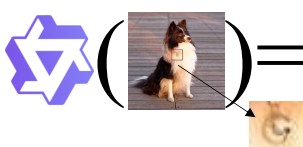 ( 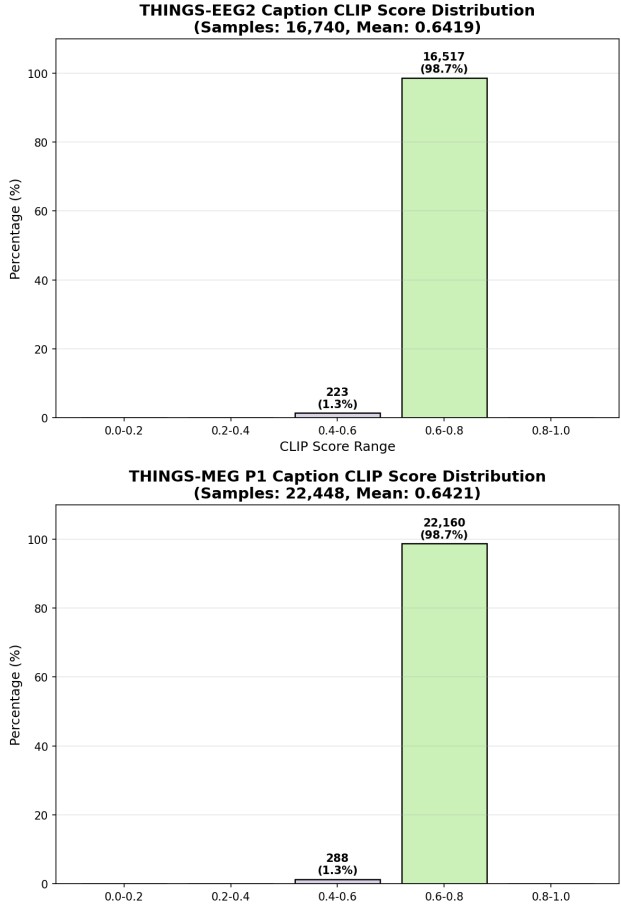 ) = {This is a photograph of a small, fluffy dog sitting on a wooden deck. The dog has a distinctive tri-color coat, with a black head and back, a white chest, and a white and brown (or tan) fluffy tail. Its ears are pointed and alert, and it is looking off to the right of the frame. The dog is wearing a small, silver-colored collar}

*Figure 4.* An example of an image caption generated by Qwen2-VL-2B-Instruct.

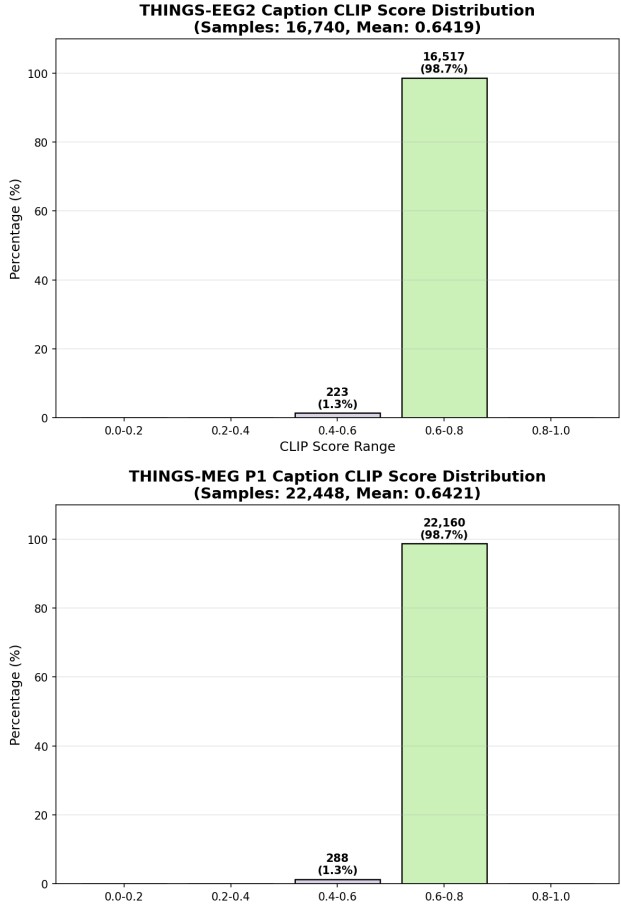

*Figure 5.* CLIP Score distributions of LLM-generated captions.

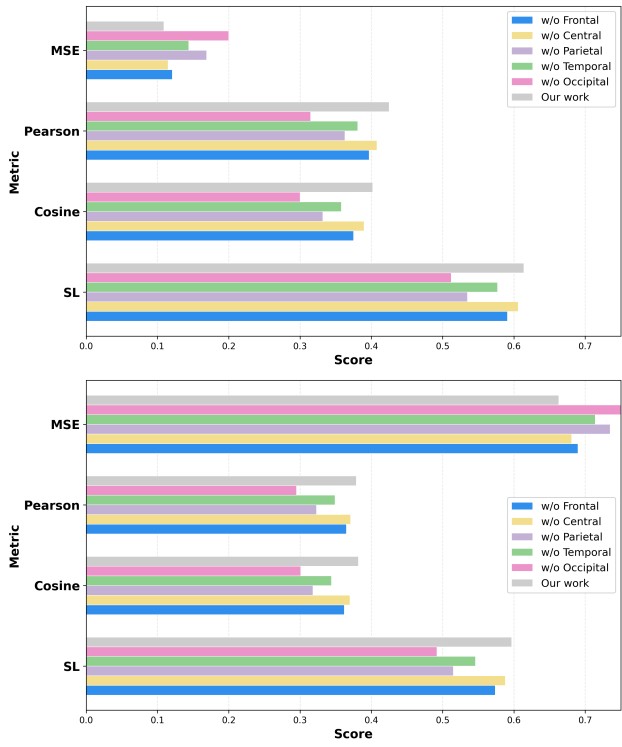

*Figure 6.* Brain region ablation study results. Top: THINGS-EEG2. Bottom: THINGS-MEG.

0.4–0.6 range, and no captions fall below 0.4. The CLIP Score distributions for THINGS-MEG Participants 2–4 are provided in Appendix I.

### 4.4. Brain Region Ablation Study

To investigate the contribution of different brain regions to brain signal generation, we conduct ablation studies by removing channels from each brain region. As shown in Figure 6, removing any brain region leads to performance degradation across all metrics. Notably, the occipital region shows the largest impact on performance. For EEG, removing the occipital region increases MSE from 0.109 to 0.200 and decreases Pearson correlation from 0.425 to 0.315. For MEG, the occipital ablation increases MSE from 0.663 to 0.759 and decreases Pearson correlation from 0.379 to 0.295. This finding aligns with neuroscience knowledge that the occipital cortex is the primary visual processing area, containing V1, V2, and V4 regions that are essential for encoding visual stimuli.

## 5. Conclusion

In this paper, we present a novel image-to-brain framework for the brain encoding stage of visual prostheses. Our framework employs a Diffusion Transformer (DiT) architecture based on DDIM to generate biologically plausible M/EEG signals from visual stimuli. To achieve effective cross-modal alignment, we design a cross-attention mechanism where brain signal embeddings serve as Query, while unified visual-semantic embeddings (concatenating CLIP

image embeddings and CLIP text embeddings from LLM-generated captions) serve as Key and Value. Furthermore, we introduce learnable spatio-temporal position embeddings that combine brain region embeddings with temporal embeddings to capture the inherent spatial and temporal characteristics of brain signals.

Extensive experiments on THINGS-EEG2 and THINGS-MEG datasets demonstrate that our method consistently outperforms existing approaches across all evaluation metrics. Ablation studies confirm the effectiveness of each proposed component, with brain region ablation analysis revealing the critical role of the occipital cortex in visual processing. Cross-subject experiments highlight the challenge of inter-individual variability in brain signals, pointing to an important direction for future research. Our work provides a foundation for advancing brain encoding in visual prostheses.

## Acknowledgements

This work was supported in part by the National Natural Science Foundation of China under Grants 62276115 and 62541606, Guangzhou Basic and Applied Basic Research Foundation under Grant 2024A04J5486, National Cyber Security-National Science and Technology Major Project under Grant 2026ZD1500303.

## Impact Statement

The positive societal impact of this work lies in its potential to improve the quality of life for blind individuals by enabling more biologically plausible brain signals in visual prosthetic devices.

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

## A. Denoising Diffusion Implicit Models (DDIM)

This appendix provides the detailed formulations of Denoising Diffusion Implicit Models (DDIM) (Song et al., 2020) used in our framework.

### A.1. Forward Process

The forward diffusion process gradually adds Gaussian noise to the original data $\mathbf{y}_0$ over $T$ timesteps. Given a noise schedule $\{\beta_t\}_{t=1}^T$, we define $\alpha_t = 1 - \beta_t$ and $\bar{\alpha}_t = \prod_{s=1}^t \alpha_s$. The forward process can be expressed as:

$$q(\mathbf{y}_t|\mathbf{y}_0) = \mathcal{N}(\mathbf{y}_t; \sqrt{\bar{\alpha}_t}\mathbf{y}_0, (1 - \bar{\alpha}_t)\mathbf{I}), \tag{1}$$

which allows direct sampling of $\mathbf{y}_t$ from $\mathbf{y}_0$:

$$\mathbf{y}_t = \sqrt{\bar{\alpha}_t}\mathbf{y}_0 + \sqrt{1 - \bar{\alpha}_t}\boldsymbol{\epsilon}, \quad \boldsymbol{\epsilon} \sim \mathcal{N}(\mathbf{0}, \mathbf{I}). \tag{2}$$

As $t$ increases, $\bar{\alpha}_t$ approaches 0, and $\mathbf{y}_T$ becomes approximately standard Gaussian noise.

### A.2. Backward Process (DDIM Sampling)

Unlike DDPM (Ho et al., 2020) which uses a Markovian reverse process, DDIM defines a non-Markovian generative process that enables faster sampling. The key insight is that the forward process marginals $q(\mathbf{y}_t|\mathbf{y}_0)$ can be matched by a family of inference distributions.

Given the predicted noise $\boldsymbol{\epsilon}_\theta(\mathbf{y}_t, t)$ from the neural network, DDIM first estimates the original data $\mathbf{y}_0$:

$$\hat{\mathbf{y}}_0 = \frac{\mathbf{y}_t - \sqrt{1 - \bar{\alpha}_t}\boldsymbol{\epsilon}_\theta(\mathbf{y}_t, t)}{\sqrt{\bar{\alpha}_t}}. \tag{3}$$

The DDIM sampling update from $\mathbf{y}_t$ to $\mathbf{y}_{t-1}$ is then given by:

$$\mathbf{y}_{t-1} = \sqrt{\bar{\alpha}_{t-1}}\hat{\mathbf{y}}_0 + \sqrt{1 - \bar{\alpha}_{t-1} - \sigma_t^2}\boldsymbol{\epsilon}_\theta(\mathbf{y}_t, t) + \sigma_t\boldsymbol{\epsilon}_t, \tag{4}$$

where $\boldsymbol{\epsilon}_t \sim \mathcal{N}(\mathbf{0}, \mathbf{I})$ is random noise and $\sigma_t$ controls the stochasticity of the sampling process.

The parameter $\sigma_t$ is defined as:

$$\sigma_t = \eta\sqrt{\frac{1 - \bar{\alpha}_{t-1}}{1 - \bar{\alpha}_t}}\sqrt{1 - \frac{\bar{\alpha}_t}{\bar{\alpha}_{t-1}}}, \tag{5}$$

where $\eta \in [0, 1]$ is a hyperparameter. When $\eta = 0$, the sampling process becomes fully deterministic, which is the setting we use in our experiments. When $\eta = 1$, the process recovers the stochasticity of DDPM.

The deterministic DDIM update ($\eta = 0$) simplifies to:

$$\mathbf{y}_{t-1} = \sqrt{\bar{\alpha}_{t-1}}\hat{\mathbf{y}}_0 + \sqrt{1 - \bar{\alpha}_{t-1}}\boldsymbol{\epsilon}_\theta(\mathbf{y}_t, t). \tag{6}$$

A key advantage of DDIM is that it allows for accelerated sampling by using a subsequence of timesteps $\{\tau_1, \tau_2, \ldots, \tau_S\} \subset \{1, 2, \ldots, T\}$ with $S \ll T$, significantly reducing the number of function evaluations required for generation while maintaining sample quality.

## B. Spatio-Temporal Position Embeddings

This appendix provides detailed descriptions of the learnable spatio-temporal position embeddings used in our framework. The position embedding for each patch is computed as the sum of a brain region embedding and a temporal embedding:

$$\mathbf{e}_{\text{pos}} = \mathbf{e}_{\text{region}} + \mathbf{e}_{\text{temporal}}$$

### B.1. Brain Region Embeddings

Brain region embeddings encode the spatial location of each patch by indicating which brain region it belongs to. We learn a region embedding matrix $\mathbf{E}_{\text{region}} \in \mathbb{R}^{R \times D}$, where $R$ is the number of brain regions and $D$ is the hidden dimension.

**Channel Reordering:** To ensure that each patch row belongs to a single brain region, we first reorder the channels by grouping them according to their brain regions. Channels belonging to the same brain region are placed adjacently. We also apply padding to each region to ensure the channel count is divisible by the patch height.

**EEG Configuration:** For EEG signals with 63 channels, we define 5 brain regions based on electrode positions:

- Frontal (22 channels): Fp1, Fp2, AF7, AF3, AFz, AF4, AF8, F7, F5, F3, F1, F2, F4, F6, F8, FC5, FC3, FC1, FCz, FC2, FC4, FC6

- Central (14 channels): C5, C3, C1, Cz, C2, C4, C6, CP5, CP3, CP1, CPz, CP2, CP4, CP6

- Temporal (10 channels): FT9, FT7, FT8, FT10, T7, T8, TP9, TP7, TP8, TP10

- Parietal (14 channels): P7, P5, P3, P1, Pz, P2, P4, P6, P8, PO7, PO3, POz, PO4, PO8

- Occipital (3 channels): O1, Oz, O2

**MEG Configuration:** For MEG signals with 271 magnetometer channels, we define 5 brain regions:

- Frontal (67 channels)

- Central (52 channels)

- Parietal (45 channels)

- Occipital (39 channels)

- Temporal (68 channels)

**Region ID Assignment:** After channel reordering, each patch row is assigned a region ID $r \in \{1, 2, \ldots, R\}$ based on its position. All patches in the same row share the same region embedding, which is retrieved from the embedding matrix $\mathbf{E}_{\text{region}} \in \mathbb{R}^{R \times D}$.

### B.2. Temporal Embeddings

Temporal embeddings encode the temporal position of each patch within the signal using a learnable embedding matrix $\mathbf{E}_{\text{temporal}} \in \mathbb{R}^{T_p \times D}$, where $T_p$ is the number of patches along the temporal dimension.

**Temporal ID Assignment:** Each patch column corresponds to a specific time window in the brain signal. All patches in the same column share the same temporal embedding indexed by $\tau \in \{1, 2, \ldots, T_p\}$.

## C. Training and Inference Algorithms

This appendix presents the pseudocode for training and inference of our image-to-brain framework.

## C.1. Training Algorithm

---

**Algorithm 1** Training Procedure

---

**Require:** Training dataset $\mathcal{D} = \{(\mathbf{x}_{\text{img}}^{(i)}, \mathbf{y}_{\text{brain}}^{(i)})\}_{i=1}^{N}$, CLIP encoder $\mathcal{E}_{\text{CLIP}}$, LLM for caption generation, DiT model $\boldsymbol{\epsilon}_\theta$, number of epochs $E$, number of diffusion timesteps $T$

**Ensure:** Trained DiT model $\boldsymbol{\epsilon}_\theta$

1: Initialize DiT model parameters $\theta$
2: Precompute captions: $\mathbf{s}^{(i)} \leftarrow \text{LLM}(\mathbf{x}_{\text{img}}^{(i)})$ for all $i$
3: **for** epoch = 1 to $E$ **do**
4:     **for** each mini-batch $\{(\mathbf{x}_{\text{img}}, \mathbf{y}_{\text{brain}}, \mathbf{s})\}$ **do**
5:         // Extract unified embeddings
6:         $\mathbf{h}_{\text{img}} \leftarrow \mathcal{E}_{\text{CLIP}}^{\text{img}}(\mathbf{x}_{\text{img}})$                 ▷ CLIP image embeddings
7:         $\mathbf{h}_{\text{txt}} \leftarrow \mathcal{E}_{\text{CLIP}}^{\text{txt}}(\mathbf{s})$                   ▷ CLIP text embeddings
8:         $\mathbf{c}_{\text{unified}} \leftarrow \text{Concat}(\mathbf{h}_{\text{img}}, \mathbf{h}_{\text{txt}})$
9:         // Forward diffusion process
10:        Sample $t \sim \text{Uniform}(\{1, \ldots, T\})$
11:        Sample $\boldsymbol{\epsilon} \sim \mathcal{N}(\mathbf{0}, \mathbf{I})$
12:        $\mathbf{y}_t \leftarrow \sqrt{\bar{\alpha}_t}\mathbf{y}_{\text{brain}} + \sqrt{1 - \bar{\alpha}_t}\boldsymbol{\epsilon}$
13:        // Predict noise and compute loss
14:        $\hat{\boldsymbol{\epsilon}} \leftarrow \boldsymbol{\epsilon}_\theta(\mathbf{y}_t, t, \mathbf{c}_{\text{unified}})$
15:        $\mathcal{L} \leftarrow \|\boldsymbol{\epsilon} - \hat{\boldsymbol{\epsilon}}\|_2^2$
16:        Update $\theta$ via gradient descent on $\mathcal{L}$
17:     **end for**
18: **end for**
19: **return** $\boldsymbol{\epsilon}_\theta$

---

## C.2. Inference Algorithm

---

**Algorithm 2** Inference Procedure (DDIM Sampling)

---

**Require:** Input image $\mathbf{x}_{\text{img}}$, trained DiT model $\boldsymbol{\epsilon}_\theta$, CLIP encoder $\mathcal{E}_{\text{CLIP}}$, LLM, number of sampling steps $S$

**Ensure:** Generated brain signal $\mathbf{y}_0$

1:  // Extract unified embeddings
2: $\mathbf{s} \leftarrow \text{LLM}(\mathbf{x}_{\text{img}})$                       ▷ Generate caption
3: $\mathbf{h}_{\text{img}} \leftarrow \mathcal{E}_{\text{CLIP}}^{\text{img}}(\mathbf{x}_{\text{img}})$
4: $\mathbf{h}_{\text{txt}} \leftarrow \mathcal{E}_{\text{CLIP}}^{\text{txt}}(\mathbf{s})$
5: $\mathbf{c}_{\text{unified}} \leftarrow \text{Concat}(\mathbf{h}_{\text{img}}, \mathbf{h}_{\text{txt}})$
6:  // DDIM sampling
7: Sample $\mathbf{y}_T \sim \mathcal{N}(\mathbf{0}, \mathbf{I})$
8: Define timestep subsequence $\{\tau_1, \tau_2, \ldots, \tau_S\}$
9: **for** $i = S$ to 1 **do**
10:     $t \leftarrow \tau_i, t' \leftarrow \tau_{i-1}$ (with $\tau_0 = 0$)
11:     $\hat{\boldsymbol{\epsilon}} \leftarrow \boldsymbol{\epsilon}_\theta(\mathbf{y}_t, t, \mathbf{c}_{\text{unified}})$
12:     $\hat{\mathbf{y}}_0 \leftarrow \frac{\mathbf{y}_t - \sqrt{1 - \bar{\alpha}_t}\hat{\boldsymbol{\epsilon}}}{\sqrt{\bar{\alpha}_t}}$
13:     $\mathbf{y}_{t'} \leftarrow \sqrt{\bar{\alpha}_{t'}}\hat{\mathbf{y}}_0 + \sqrt{1 - \bar{\alpha}_{t'}}\hat{\boldsymbol{\epsilon}}$
14: **end for**
15: **return** $\mathbf{y}_0$

---

# D. Datasets and Preprocessing

**THINGS-EEG2 Dataset:** We conduct our experiments on the THINGS-EEG2 dataset (Gifford et al., 2022), which represents one of the largest and most diverse EEG-image paired datasets currently available. This dataset employs a rapid

serial visual presentation (RSVP) paradigm and contains EEG recordings from ten participants. The training set comprises 1,654 concepts × 10 images × 4 repetitions, while the test set includes 200 concepts × 1 image × 80 repetitions. Each image is presented for 100 ms followed by a 100 ms blank screen, with a stimulus onset asynchrony (SOA) of 200 ms. The data were recorded using 63 electrode channels at a sampling rate of 1000 Hz with bandpass filtering at [0.1, 100] Hz. For preprocessing, we select all 63 electrode channels and perform epoching from -200 ms to 1000 ms relative to stimulus onset, followed by baseline correction using the pre-stimulus period. The data is then downsampled to 250 Hz and whitened using Multivariate Noise Normalization (MVNN) (Guggenmos et al., 2018), which computes covariance matrices across epochs and applies whitening transformation. The final EEG data has dimensions of 63 channels × 250 time points (0–1000 ms at 250 Hz).

**THINGS-MEG Dataset:** We also evaluate our framework on the THINGS-MEG dataset (Hebart et al., 2023) containing four participants and paired MEG recordings with corresponding visual stimuli. This dataset offers better spatial resolution and more stable responses with a longer SOA of $1500 \pm 200$ ms, including a 500-ms stimulus followed by a jitter blank screen. The training stage includes 1,854 concepts × 12 images × 1 repetition, while the test stage includes 200 concepts × 1 image × 12 repetitions. The data were recorded using 271 magnetometer channels and filtered to [0.1, 100] Hz. For preprocessing, we exclude one faulty channel and retain 271 magnetometer channels. The data is bandpass filtered at [0.1, 100] Hz, epoched from -100 ms to 1300 ms, and baseline corrected using z-score normalization. We then crop the data to the 0–1000 ms post-stimulus window and downsample to 200 Hz. The final MEG data has dimensions of 271 channels × 200 time points (0–1000 ms at 200 Hz).

## E. Experiment Details

We implement our framework using PyTorch and Accelerate for distributed training on four NVIDIA V100S GPUs. We use the AdamW optimizer with a learning rate of $1 \times 10^{-4}$ and weight decay of $1 \times 10^{-5}$. For batch size, we use 16 for THINGS-EEG2 and 4 for THINGS-MEG due to memory constraints from the higher dimensional MEG data.

**Model Architecture:** Our DiT model uses a hidden dimension of 768, 12 transformer layers, 12 attention heads, and MLP ratio of 4.0. The patch size is set to $(4, 4)$ for both EEG and MEG data. For EEG signals with dimensions of $(63, 250)$, the model processes $16 \times 63 = 1008$ patches. For MEG signals with dimensions of $(271, 200)$, the model processes $68 \times 50 = 3400$ patches. We use the ViT-L/14 variant of CLIP as our visual encoder, which produces patch-level features with 257 tokens (1 CLS token + 256 patch tokens) of dimension 768. The CLIP text encoder is used to encode the LLM-generated captions, producing text embeddings with 77 tokens of dimension 768. The diffusion process uses 1000 training timesteps with a linear noise schedule.

**Leave-One-Subject-Out (LOSO) Protocol:** To further investigate cross-subject generalization, we adopt a leave-one-subject-out protocol. Taking THINGS-EEG2 as an example: for each fold, we hold out one subject and combine the training data of the remaining 9 subjects into a unified training set, with their test data forming a unified test set. A single model is trained on this combined training set and then evaluated in two ways: (1) on the combined test set of the 9 training subjects, yielding the "Within-subject" result, and (2) on the test data of the held-out subject, yielding the "Cross-subject" result. Averaging across all 10 folds produces the final leave-one-subject-out results. The same protocol is applied to THINGS-MEG with 4 subjects.

## F. Evaluation Metrics

This appendix provides detailed definitions of the evaluation metrics used to assess the quality of generated brain signals.

**Mean Squared Error (MSE):** MSE measures the average squared difference between the generated and ground truth brain signals across all channels and time points. It quantifies the magnitude of reconstruction error, where lower values indicate that the generated signal is closer to the ground truth in terms of absolute amplitude.

**Pearson Correlation Coefficient:** The Pearson Correlation Coefficient measures the linear relationship between the generated and ground truth signals. It captures how well the temporal dynamics and patterns of the generated signal match those of the ground truth, regardless of differences in scale or offset. Values range from -1 to 1, with higher values indicating stronger positive correlation.

**Cosine Similarity:** Cosine Similarity measures the angular similarity between the generated and ground truth signals when treated as high-dimensional vectors. It evaluates whether the overall pattern and direction of the generated signal align with

the ground truth, independent of their magnitudes. Values range from -1 to 1, with higher values indicating better pattern alignment.

**Synchronization Likelihood (SL):** Synchronization Likelihood (Stam & Van Dijk, 2002) is a nonlinear measure that quantifies the synchronization between two time series. It is particularly suitable for evaluating the biological plausibility of generated brain signals as it captures nonlinear dependencies. For two time series $X$ and $Y$, SL is computed based on the probability that the recurrence of a pattern in $X$ coincides with a recurrence in $Y$. The computation involves: (1) constructing delay embedding vectors for both time series with embedding dimension $m$ and delay $\tau$; (2) for each time point $i$, finding the $k$ nearest neighbors in the embedded space; (3) computing the probability that when $X$ has a recurrence (neighbor), $Y$ also has a recurrence at the same time. SL ranges from 0 to 1, where higher values indicate stronger synchronization between the generated and ground truth signals, suggesting that the generated signal preserves the temporal dynamics of real brain activity.

## G. Baseline Methods

This appendix provides detailed descriptions of the baseline methods used in our experiments.

### G.1. Traditional Encoding Models

**Güçlü *et al.* (Güçlü & Van Gerven, 2015):** This work investigates how deep neural networks can be used to understand neural representations in the ventral visual stream. The authors use a pre-trained Convolutional Neural Network (CNN) to extract nonlinear feature representations from images and employ regularized linear regression to predict brain signals. Their key finding is that CNN layers exhibit a gradient of complexity that mirrors the hierarchy of the ventral stream, with early layers corresponding to V1 and deeper layers to downstream areas such as LO.

**Yamins *et al.* (Yamins et al., 2014):** This work demonstrates that hierarchical neural network models optimized for object categorization can accurately predict neural responses in the primate ventral visual stream. The authors show that the top output layer of hierarchical CNN models optimized for categorization is highly predictive of IT neural responses, while intermediate layers are highly predictive of V4 neural responses.

### G.2. Generative Approaches

**UNet-based Diffusion:** To validate the advantage of the transformer-based architecture, we implement a UNet-based Diffusion baseline that replaces our DiT backbone with a conventional U-Net architecture (Ronneberger et al., 2015) while keeping all other components (DDIM framework, unified visual-semantic embeddings, spatio-temporal position embeddings) identical.

**MindSimulator (Bao et al., 2025):** MindSimulator is a recent brain signal generation framework that employs a latent diffusion model for synthesizing neural responses from visual stimuli. It uses a variational autoencoder to compress brain signals into a latent space and performs diffusion-based generation in this compressed representation, enabling efficient synthesis of high-dimensional brain data.

**SynBrain (Mai et al., 2025):** SynBrain (Mai et al., 2025) is a generative framework designed to enhance visual-to-fMRI synthesis through probabilistic representation learning. The method aims to simulate the transformation from visual semantics to neural responses while capturing biological variability and maintaining functional consistency.

SynBrain consists of two key components: (1) **BrainVAE**, which models neural representations as continuous probability distributions through probabilistic learning, using visual semantic constraints to maintain functional consistency; (2) **Semantic-to-Neural Mapper**, which serves as a semantic transfer pathway to project visual semantics onto the neural response manifold for high-fidelity fMRI signal synthesis.

While SynBrain was originally designed for fMRI synthesis, we adapt it as a baseline for M/EEG generation in our experiments. The key difference between our method and SynBrain lies in the architecture: we employ a Diffusion Transformer (DiT) with cross-attention mechanisms for cross-modal alignment, whereas SynBrain uses a VAE-based approach with a semantic mapper. Additionally, our method incorporates LLM-generated captions and learnable spatio-temporal position embeddings specifically designed for M/EEG signals.

## H. Cross-subject Topography for THINGS-MEG

This appendix provides the MEG cross-subject topography comparison, supplementing the EEG results presented in Figure 3. Figure 7 compares the topographies of Participant 1's training data with Participant 2's test data for THINGS-MEG. Similar to the EEG results, the difference topographies reveal substantial inter-individual variability across all time windows, explaining the performance degradation observed in cross-subject settings for MEG signals.

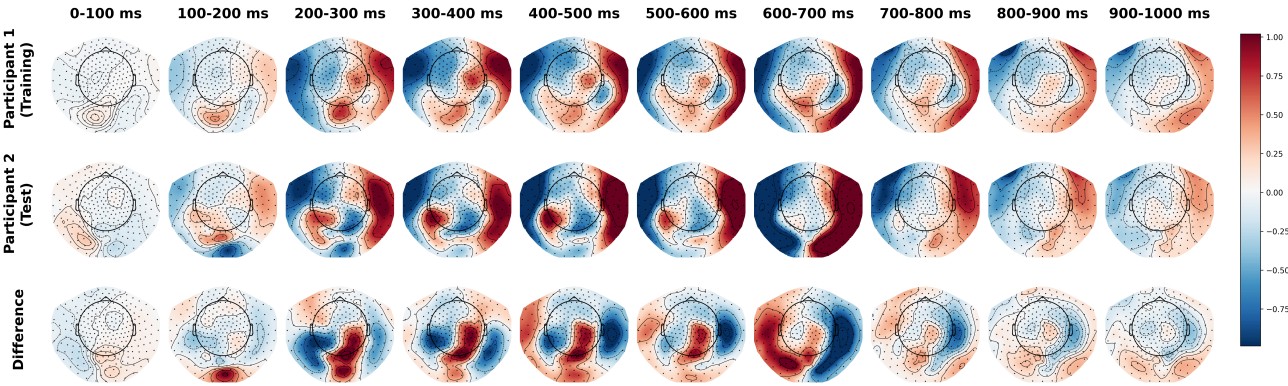

*Figure 7.* Cross-subject topography comparison for THINGS-MEG. The figure shows MEG topographies comparing Participant 1 (training) and Participant 2 (test). The third row shows the difference between training and test participants, highlighting inter-individual variability in brain signals.

## I. Supplementary Results for Caption Quality Analysis

This appendix provides supplementary results for the LLM-Generated Caption Quality Analysis presented in Section 4.3. We present two types of additional results: (1) CLIP Score distributions for THINGS-MEG Participants 2–4 using Qwen2-VL-2B-Instruct, and (2) CLIP Score distributions for captions generated by the baseline LLMs (VisualGLM and MiniGPT-4).

**CLIP Score Distributions for THINGS-MEG Participants 2–4:** Figure 8 shows the CLIP Score distributions for THINGS-MEG Participants 2–4 using Qwen2-VL-2B-Instruct. All three participants exhibit similar distribution patterns to Participant 1, with mean CLIP Scores around 0.642 and over 98% of captions falling in the 0.6–0.8 range, demonstrating the consistent quality of captions generated by Qwen2-VL-2B-Instruct across different participants.

**CLIP Score Distributions for Baseline LLMs:** Tables 7 and 8 present the CLIP Score distributions for captions generated by VisualGLM and MiniGPT-4, respectively. For VisualGLM, approximately 37% of captions fall in the 0.4–0.6 range, while 63% fall in the 0.6–0.8 range. For MiniGPT-4, the distribution shifts toward higher scores, with approximately 27% in the 0.4–0.6 range and 73% in the 0.6–0.8 range. Compared to Qwen2-VL-2B-Instruct (with only 1.3% in the 0.4–0.6 range and 98.7% in the 0.6–0.8 range), both baseline LLMs produce captions with lower semantic alignment to the original images, which explains the performance gap observed in Tables 5 and 6.

*Table 7.* CLIP Score distribution of captions generated by VisualGLM

| Dataset | CLIP Score Range | | | | |
|---|---|---|---|---|---|
| | $[0.0, 0.2)$ | $[0.2, 0.4)$ | $[0.4, 0.6)$ | $[0.6, 0.8)$ | $[0.8, 1.0]$ |
| THINGS-EEG | 0.0 | 0.0 | 37.5%(6, 281) | 62.5%(10, 459) | 0.0 |
| THINGS-MEG P1 | 0.0 | 0.0 | 36.9%(8, 281) | 63.1%(14, 167) | 0.0 |
| THINGS-MEG P2 | 0.0 | 0.0 | 37.3%(8, 368) | 62.7%(14, 080) | 0.0 |
| THINGS-MEG P3 | 0.0 | 0.0 | 37.0%(8, 312) | 63.0%(14, 136) | 0.0 |
| THINGS-MEG P4 | 0.0 | 0.0 | 37.3%(8, 369) | 62.7%(14, 079) | 0.0 |

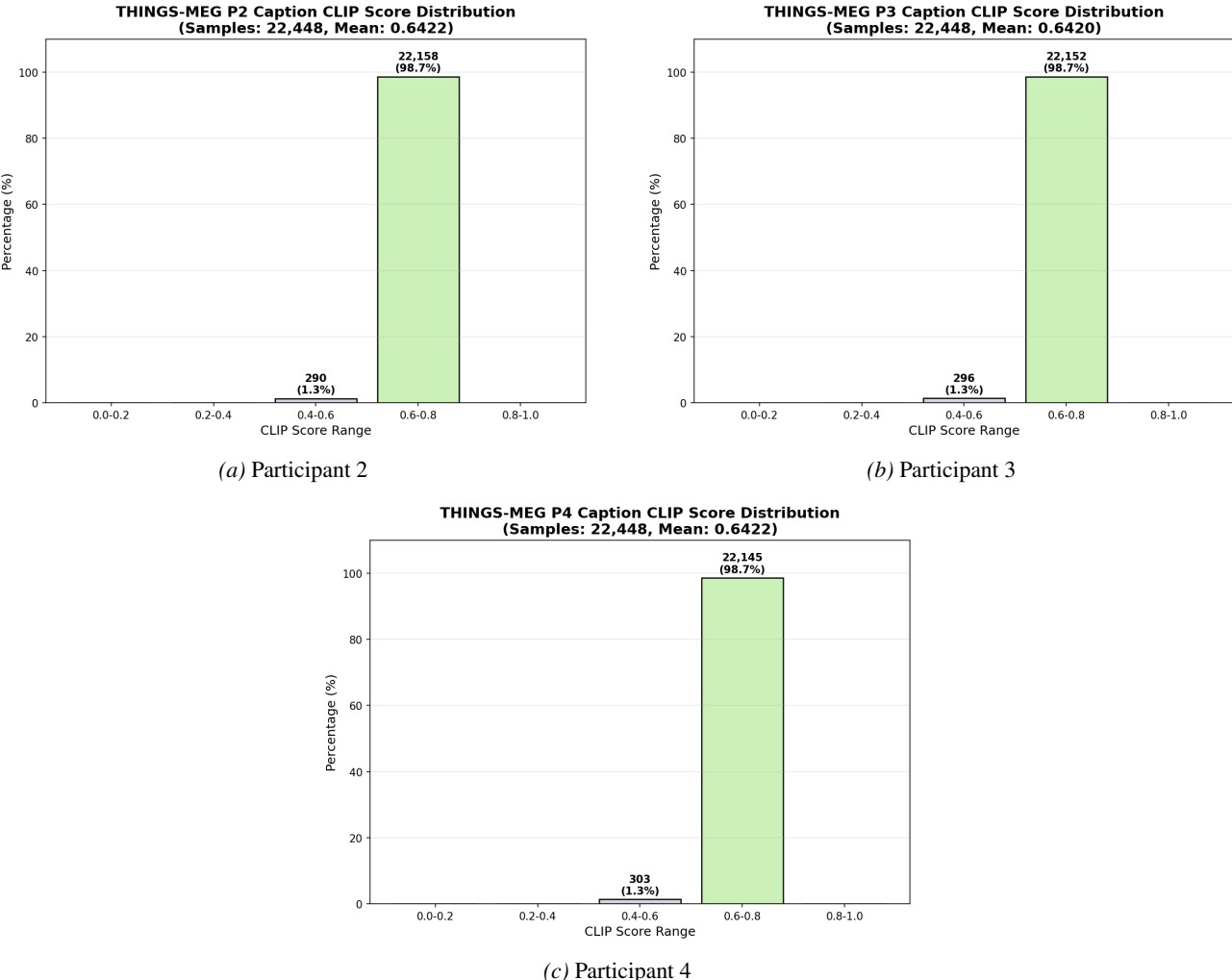

*(a)* Participant 2
*(b)* Participant 3

*(c)* Participant 4

*Figure 8.* CLIP Score distributions for THINGS-MEG Participants 2–4 using Qwen2-VL-2B-Instruct.

*Table 8.* CLIP Score distribution of captions generated by MiniGPT-4

| Dataset | CLIP Score Range | | | | |
|---|---|---|---|---|---|
| | $[0.0, 0.2)$ | $[0.2, 0.4)$ | $[0.4, 0.6)$ | $[0.6, 0.8)$ | $[0.8, 1.0]$ |
| THINGS-EEG | 0.0 | 0.0 | $26.6\%(4,459)$ | $73.4\%(12,281)$ | 0.0 |
| THINGS-MEG P1 | 0.0 | 0.0 | $27.1\%(6,092)$ | $72.9\%(16,356)$ | 0.0 |
| THINGS-MEG P2 | 0.0 | 0.0 | $27.3\%(6,118)$ | $72.7\%(16,330)$ | 0.0 |
| THINGS-MEG P3 | 0.0 | 0.0 | $26.7\%(5,984)$ | $73.3\%(16,464)$ | 0.0 |
| THINGS-MEG P4 | 0.0 | 0.0 | $26.9\%(6,044)$ | $73.1\%(16,404)$ | 0.0 |

