# OpenReview forum: "Image-to-Brain Signal Generation for Visual Prosthesis with CLIP Guided Multimodal Diffusion Models"
_ICML.cc/2026/Conference — ICML 2026 regular_

### Official Review · Reviewer_NPdM · 2026-03-11

**Soundness:** 3
**Presentation:** 2
**Significance:** 3
**Originality:** 2
**Overall Recommendation:** 4
**Confidence:** 1

**Summary:**

This submission investigates the relatively uncharted domain of image-to-brain encoding for visual prostheses, shifting focus from decoding neural activity to synthesizing M/EEG responses directly from visual stimuli. The proposed approach utilizes a DiT/DDIM-style diffusion framework, where generation is conditioned on combined CLIP image features and CLIP text features derived from LLM-generated captions. To accurately capture the intricate spatio-temporal dynamics inherent in EEG and MEG signals, the model incorporates learnable, region-aware positional embeddings alongside temporal positional embeddings. Evaluated on the THINGS-EEG2 and THINGS-MEG datasets using within-subject protocols, the method demonstrably surpasses traditional encoding baselines as well as the generative baseline SynBrain. Finally, the paper provides comprehensive ablation studies analyzing the impact of text conditioning, specific embedding strategies, and the removal of brain regions.

**Compliance With Llm Reviewing Policy:**

Affirmed.

**Final Justification:**

The authors have addressed most of my concerns, so I am raising my score to 4.

**Key Questions For Authors:**

Can the authors provide stronger evidence that the reported signal-similarity gains correspond to functionally meaningful improvements, rather than merely better matching low-level statistics of the recorded signals?



Why were these specific baselines chosen? Are there more recent neural encoding or conditional generative baselines that would offer a tougher comparison?

**Limitations:**

yes

**Strengths And Weaknesses:**

Strengths:

1. While the majority of contemporary research prioritizes brain decoding, this manuscript shifts the focus to the reverse process of mapping images to M/EEG signals, an approach that is highly applicable to both neural encoding modeling and the development of visual prostheses. From a conceptual standpoint, this specific design aligns effectively with the underlying structure of neurophysiological data.

2. By assessing performance on MEG and EEG datasets, comparing the approach to both traditional encoding methods and a modern generative baseline, and conducting ablation studies, the authors successfully demonstrate the essential roles of text, temporal, and region embeddings.

3. The inclusion of analyses oriented towards neuroscientific principles adds valuable depth to the research. Specifically, demonstrating that excluding the occipital region causes the greatest performance drop aligns with biological expectations, grounding the model in reality rather than treating it merely as an opaque computational exercise.

Weaknesses:

1. Cross-subject generalization is weak. The within-subject numbers are clearly better than baselines, but cross-subject performance drops sharply on both datasets, especially on THINGS-MEG, which raises concerns about robustness and clinical practicality.

2. The baseline set may not be strong enough. Two baselines are old CNN-based encoding models, and only one recent generative baseline is included. It is not yet clear whether the gains come from diffusion itself, stronger conditioning, more capacity, or simply more modern feature extractors.

3. The paper shows that removing text embeddings hurts and that Qwen2-VL captions outperform two alternative multimodal LLMs under the chosen metrics, but this does not fully disentangle whether the gain comes from richer semantics, model size differences, prompt style, or CLIP-text compatibility.

---

> ### Author Rebuttal · Authors · 2026-03-30
>
> We thank the reviewer for the thoughtful and constructive questions. We address each one in detail below.
>
> **Q1: Evidence that signal-similarity gains correspond to functionally meaningful improvements.**
>
> This is an excellent and important question. We agree that signal-level metrics alone (MSE, Pearson, Cosine) may not fully capture functional meaningfulness. However, we note that the Synchronization Likelihood (SL) metric, already included in our evaluation, is a nonlinear measure that goes beyond low-level statistics. SL captures the preservation of temporal dynamics and nonlinear synchronization patterns between generated and real signals, which are functionally relevant properties of brain activity (Stam & van Dijk, 2002). Our method's improvement in SL (e.g., 0.577 → 0.614 on EEG and 0.514 → 0.597 on MEG compared to SynBrain) suggests that the gains are not limited to low-level signal similarity, but also reflect the preservation of functionally relevant nonlinear neural dynamics.
>
>
> **Q2 & W1 & W2: Baseline choices, stronger baselines, and cross-subject generalization.**
>
> We acknowledge the reviewer's concerns regarding baseline strength and cross-subject generalization. To address these, we conduct additional experiments with two new baselines: (1) a UNet-based Diffusion model, which replaces DiT with a UNet backbone; (2) "MindSimulator: Exploring Brain Concept Localization via Synthetic FMRI" (Bao et al., 2025), a recent generative encoding method. We also adopt a leave-one-subject-out (LOSO) training strategy to evaluate our method's performance with larger data volumes and its generalization ability on unseen subjects when the training set includes multiple subjects. Due to time constraints, we currently report LOSO results on THINGS-MEG only; results on THINGS-EEG2 will be provided in the next round of discussion. Results are shown below:
>
> **Table 1: Quantitative evaluation on THINGS-EEG2**
>
> | Method | MSE ↓ | Pearson ↑ | Cosine ↑ | SL ↑ |
> |--------|-------|-----------|----------|------|
> | Güçlü et al. (2015) | 0.211 | 0.178 | 0.181 | 0.458 |
> | Yamins et al. (2014) | 0.196 | 0.203 | 0.209 | 0.476 |
> | UNet-based Diffusion | 0.217 | 0.188 | 0.185 | 0.471 |
> | MindSimulator (Bao et al., 2025) | 0.160 | 0.349 | 0.345 | 0.543 |
> | SynBrain (Mai et al., 2025) | 0.156 | 0.366 | 0.347 | 0.577 |
> | **Our work** | **0.109** | **0.425** | **0.402** | **0.614** |
> | Our work (Cross-subject) | 0.174 | 0.157 | 0.165 | 0.522 |
>
> **Table 2: Quantitative evaluation on THINGS-MEG**
>
> | Method | MSE ↓ | Pearson ↑ | Cosine ↑ | SL ↑ |
> |--------|-------|-----------|----------|------|
> | Güçlü et al. (2015) | 0.831 | 0.243 | 0.254 | 0.405 |
> | Yamins et al. (2014) | 0.796 | 0.269 | 0.274 | 0.423 |
> | UNet-based Diffusion | 0.763 | 0.122 | 0.127 | 0.389 |
> | MindSimulator (Bao et al., 2025) | 0.755 | 0.281 | 0.274 | 0.472 |
> | SynBrain (Mai et al., 2025) | 0.727 | 0.302 | 0.326 | 0.514 |
> | **Our work** | **0.663** | **0.379** | **0.382** | **0.597** |
> | LOSO (Within-subject) | 1.015 | 0.237 | 0.215 | 0.437 |
> | Our work (Cross-subject) | 0.768 | 0.082 | 0.089 | 0.461 |
> | LOSO (Cross-subject) | 1.237 | 0.164 | 0.157 | 0.411 |
>
> To address Reviewer jCbw's Q3, Reviewer HfVL's W2, and Reviewer NPdM's Q2 & W2, we add two new baselines for comparison: the UNet-based Diffusion model and MindSimulator (Bao et al., 2025). Our method outperforms both baselines on all datasets and across all metrics.
>
> The LOSO results on THINGS-MEG are provided to address Reviewer HfVL's W4 and Reviewer NPdM's W1. We observe that the LOSO cross-subject results show notable improvement over the per-subject cross-subject (Our work (Cross-subject)) results, with Pearson improving from 0.082 to 0.164 and Cosine from 0.089 to 0.157, indicating that incorporating data from multiple subjects effectively enhances cross-subject generalization. However, we also observe that the LOSO within-subject results are lower than the per-subject within-subject (Our work) results, suggesting the need to explore models that are more robust to larger data volumes.

---

> > ### Author Rebuttal · Reviewer_NPdM · 2026-04-03
> >
> > The authors have addressed most of my concerns, so I am raising my score to 4.

---

> > > ### Author Response · Authors · 2026-04-04
> > >
> > > **Table 1: Quantitative evaluation on THINGS-EEG2**
> > >
> > > | Method | MSE ↓ | Pearson ↑ | Cosine ↑ | SL ↑ |
> > > |--------|-------|-----------|----------|------|
> > > | Güçlü et al. (2015) | 0.211 | 0.178 | 0.181 | 0.458 |
> > > | Yamins et al. (2014) | 0.196 | 0.203 | 0.209 | 0.476 |
> > > | UNet-based Diffusion | 0.217 | 0.188 | 0.185 | 0.471 |
> > > | MindSimulator (Bao et al., 2025) | 0.160 | 0.349 | 0.345 | 0.543 |
> > > | SynBrain (Mai et al., 2025) | 0.156 | 0.366 | 0.347 | 0.577 |
> > > | **Our work** | **0.109** | **0.425** | **0.402** | **0.614** |
> > > | LOSO (Within-subject) | 0.200 | 0.293 | 0.307 | 0.535 |
> > > | Our work (Cross-subject) | 0.174 | 0.157 | 0.165 | 0.522 |
> > > | LOSO (Cross-subject) | 0.285 | 0.209 | 0.220 | 0.493 |
> > >
> > >
> > > We sincerely thank the reviewer for the positive acknowledgement and for raising the score. We truly appreciate the time and effort invested in providing constructive feedback, which has significantly strengthened our work.
> > >
> > > We would like to provide an update on the LOSO experiments. The THINGS-EEG2 LOSO results are now available and presented in Table 1. The findings on THINGS-EEG2 are consistent with those on THINGS-MEG: the LOSO cross-subject results show improvement over the per-subject cross-subject (Our work (Cross-subject)) results on Pearson (0.157 → 0.209) and Cosine (0.165 → 0.220), confirming that incorporating data from multiple subjects effectively enhances cross-subject generalization across both modalities. Meanwhile, the LOSO within-subject results remain lower than the per-subject within-subject results, again consistent with the THINGS-MEG observations.
> > >
> > > To further understand why the LOSO cross-subject model outperforms "Our work (Cross-subject)", we visualize the topographies of all 10 subjects in THINGS-EEG2 and all 4 subjects in THINGS-MEG. Due to the large size of the figures, these topographies are not included in the manuscript, but we provide a summary of the key observations here.
> > > The topographies confirm that the neural responses are consistent with the bottom-up hierarchy of the visual system, where visual stimuli are processed sequentially by V1, V2, V4 in the occipital cortex, and the inferotemporal cortex in the temporal cortex along the ventral stream for object recognition. However, while the response locations remain roughly consistent (primarily in the occipital and temporal cortex regions), the spatial extent and amplitude magnitudes exhibit substantial variations.
> > > This observation provides a direct explanation for the advantage of LOSO training in the cross-subject setting: by training on data from multiple subjects simultaneously, the model is exposed to a diverse range of spatial distributions and amplitude patterns, effectively learning to accommodate inter-individual variability.
> > > In contrast, "Our work (Cross-subject)" is trained on only a single subject's data, limiting its capacity to generalize to unseen subjects with different topographic characteristics.

---

### Official Review · Reviewer_Vseb · 2026-03-11

**Soundness:** 3
**Presentation:** 3
**Significance:** 3
**Originality:** 3
**Overall Recommendation:** 4
**Confidence:** 1

**Summary:**

This paper introduces a image-to-brain signal generation framework, which is tailored for the brain encoding stage of visual prostheses. Unlike most existing work that decodes brain activity into images, this study addresses the underexplored inverse problem: translating images into biologically plausible M/EEG signals. The proposed approach leverages a DiT model enhanced with cross attention mechanism. Cross attention dynamically aligns CLIP-extracted visual and textual embeddings with brain signal representations. Additionally, the model utilizes learnable spatio-temporal position encodings to accurately capture the spatial and temporal dynamics of neural responses .

**Compliance With Llm Reviewing Policy:**

Affirmed.

**Key Questions For Authors:**

I have very little expertise in this area, but I want to ask whether and how this work could be applied to real, clinical cases. If the applicability is demonstrated by the authors, I will raise my score.

**Limitations:**

Yes

**Strengths And Weaknesses:**

Strengths:

1. This paper effectively addresses the underexplored image-to-brain encoding problem using multimodal diffusion.

2. In general, this paper is easy to follow.

Weaknesses:

1. The heavy reliance on LLM-generated image captions to guide the CLIP text embeddings could introduce semantic hallucinations.

2. M/EGG datasets are used, but it remains unclear if the method is applicable to clinical cases which might be different from M/EGG.

---

> ### Author Rebuttal · Authors · 2026-03-30
>
> We sincerely thank the reviewer for this important question and the encouraging feedback.
>
> **W1: LLM-generated captions could introduce semantic hallucinations.**
>
> We appreciate this concern. Our framework already incorporates several design choices that inherently mitigate this risk:
>
> (1) We use Qwen2-VL-2B-Instruct, which is specifically designed for visual understanding tasks and generates captions grounded in the actual image content.
>
> (2) As reported in our paper, the mean CLIP Score between images and generated captions is 0.6419 for THINGS-EEG2 and 0.6421 for THINGS-MEG, with 98.7% of captions achieving scores in the 0.6–0.8 range and no captions falling below 0.4, indicating high semantic fidelity.
>
> (3) CLIP text embeddings serve as a complementary signal alongside the CLIP image embeddings (which are hallucination-free), so the model is not solely dependent on the captions.
>
> **W2 & Q1: Applicability to clinical cases beyond M/EEG datasets and clinical applicability of the proposed framework.**
>
> A recent clinical study published in the New England Journal of Medicine, "Subretinal Photovoltaic Implant to Restore Vision in Geographic Atrophy Due to AMD" (Holz et al., NEJM 2026), provides a detailed description of the visual prosthesis workflow based on real clinical cases and demonstrates its clinical applicability. The Photovoltaic Retina Implant Microarray (PRIMA) system combines a subretinal photovoltaic implant, a pocket processor, and glasses that project near-infrared light to the implant to restore sight to areas of central retinal atrophy. A camera on the glasses captures visual scenes and sends the information to the processor. The processed information is projected onto the subretinal implant via near-infrared light (880 nm). Each pixel in the implant contains photodiodes that convert the near-infrared light into electrical current to stimulate retinal bipolar cells. This electrical information is then processed in the retina and transferred via the optic nerve to the brain, where it is interpreted as a visual image. In this clinical trial involving 38 participants with geographic atrophy due to AMD, the PRIMA system leads to a clinically meaningful improvement in visual acuity in 81% of participants at 12 months.
>
> However, a key limitation of the PRIMA system is that its photodiodes perform a linear photoelectric conversion from light to electrical signals, as described in "Physics of Semiconductor Devices" (Sze, Li & Ng, 2021). In contrast, the mapping from visual images to neural responses in the brain is inherently nonlinear (as demonstrated in "Jacobian-Based Interpretation of Nonlinear Neural Encoding Model"). Our proposed framework aims to replace the role of photodiodes by learning this nonlinear image-to-brain-signal mapping, thereby enabling a more biologically plausible conversion from visual stimuli to electrical stimulation patterns. This could potentially improve the quality of restored vision in future visual prosthesis systems.
>
> Moreover, our framework is designed to be agnostic to the type of brain signal. The core components — the diffusion process, cross-attention conditioning on CLIP embeddings, and the denoising objective — are independent of the specific neural signal type. The brain-signal-specific components are limited to the patchification strategy and the spatio-temporal position encoding. For example, in this paper, EEG signals have dimensions of [batch_size, 1, 63, 250] and MEG signals have dimensions of [batch_size, 1, 271, 200]. The difference between them lies only in the patchification configuration (patch size and number of patches) and the temporal resolution in the spatio-temporal position encoding, while the brain region partition, diffusion process, and cross-attention conditioning remain identical. Adapting to other types of neural responses follows the same principle.

---

> > ### Author Rebuttal · Reviewer_Vseb · 2026-04-05
> >
> > Thank you very much for your response!

---

### Official Review · Reviewer_HfVL · 2026-03-13

**Soundness:** 2
**Presentation:** 3
**Significance:** 2
**Originality:** 2
**Overall Recommendation:** 2
**Confidence:** 3

**Summary:**

The paper proposes a DiT-based generative model for the Image-to-Brain M/EEG encoding task. The experiments on THINGS-EEG2 and THINGS-MEG benchmarks show that the proposed model can work well when trained subject by subject.

**Compliance With Llm Reviewing Policy:**

Affirmed.

**Final Justification:**

I appreciate the authors’ rebuttal and follow-up. However, the concern regarding the use of DDIM, a generative model, for brain encoding has not been fully resolved. The key issue is that the authors fail to provide evidence showing whether their modeling of distribution noise is related to non-stimulus-related factors or to stimulus-related factors. If it is the latter, I would acknowledge this as a significant improvement for AI in neuroscience. Nevertheless, I remain skeptical about whether generative diffusion models should be used for brain encoding at all. Therefore, I have decided to keep my score unchanged.

**Key Questions For Authors:**

See weaknesses

**Limitations:**

yes

**Strengths And Weaknesses:**

## Strengths
1. The method section and technical details of the proposed method are clearly explained
2. The introduced temporal and regional embedding considers the traits of M/EEG modality and is well motivated
3. The experiments on THINGS-EEG2 and THINGS-MEG show that the proposed approach achieves the SOTA performance across four metrics.

## Weaknesses
1. The authors frame the image-to-brain neural encoding task as a generative process using a Diffusion-based framework. However, the fundamental nature of neural encoding is a deterministic mapping (or a constrained predictive task) where neural responses are strictly stimulus-dependent and subject-specific. This raises significant concerns regarding the biological and functional relevance of the chosen model. More specifically, Diffusion models are designed to map Gaussian noise to a diverse manifold of images. In neural encoding, **generation from noise** lacks a biological justification. In addition, do we really seek **diversity** in predicted brain activity for a fixed stimulus? If yes, the paper definitely lacks a rigorous justification for why a generative probabilistic model (Diffusion) is superior to a high-capacity discriminative model, such as a Transformer-based predictor. Therefore, the narrative should be revised to explicitly address what information a standard regression or Transformer model fails to capture that Diffusion does. If the argument is about capturing the underlying distribution of neural noise, this needs to be empirically validated.
2. The justification for choosing DiT over the UNet-based diffusion model is "the DiT architecture offers better scalability and has demonstrated superior performance in generative tasks." This is true in CV or NLP, but to my understanding and knowledge, it has not been validated or proved in neuroscience-related research. To improve the manuscript, the author should explain more about the specific advantage of using DiT over classic diffusion models in neural encoding.
3. While the adaptation of high-performance architectures like DiT to neuroscience applications is a valid research direction, a "plug-and-play" approach lacks the depth required for a top-tier machine learning contribution. The current novel implementation merely adds regional and temporal embeddings to M/EEG features, which constitutes a marginal architectural adjustment rather than a principled integration of neural constraints.
4. The quantitative results in Tables 1 and 2 reveal a significant performance gap between per-subject and cross-subject models. While the transparency regarding these limitations is appreciated and noted by the authors, the lack of subject generalizability undermines the practical utility of the proposed architecture. This suggests that the current DiT adaptation fails to capture a universal latent space for neural representations across different individuals, although this may not be the focus of the paper. However, the poor cross-subject performance indicates an opportunity to move beyond an "all-in-one" architecture. To reach the standard expected for a top-tier publication, the authors should consider incorporating subject-specific adapters or neural alignment layers. A model that successfully leverages cross-subject data while accounting for individual variability would increase biological plausibility. In addition, large-scale models like DiT require massive datasets for training, meaning that a robust cross-subject framework would allow the model to finally benefit from the aggregate data of multiple participants, which is currently a missed opportunity in this manuscript.

---

> ### Author Rebuttal · Authors · 2026-03-30
>
> We greatly appreciate the reviewer's thorough and constructive critique. We address each concern raised in the weaknesses below.
>
> **W1: Why use a diffusion model for neural encoding? Is a generative model justified over a discriminative one?**
>
> This is an excellent and fundamental question. We provide both theoretical justification and empirical evidence:
>
> (a) Neural responses are inherently stochastic, not deterministic. As also noted by our baseline SynBrain (Mai et al., 2025), one fundamental characteristic of brain activity is that the mapping from visual stimuli to neural responses is inherently one-to-many, influenced by trial-level noise, brain attentional fluctuations, and inter-individual variability. This means that neural encoding is not a deterministic mapping but a probabilistic process, which naturally motivates the use of generative models.
>
> (b) The datasets themselves demonstrate this stochasticity. In THINGS-EEG2, each test image is presented 80 times to the same subject, yielding 80 distinct EEG responses. In THINGS-MEG, each test image is presented 12 times. If neural encoding were truly deterministic, these repetitions would be identical — but they are not. A generative probabilistic model like diffusion is naturally suited to capture this response distribution $p(\mathbf{y}|\mathbf{x})$, whereas a deterministic regression model can only predict the conditional mean $\mathbb{E}[\mathbf{y}|\mathbf{x}]$, losing information about the response variability.
>
> (c) DDIM deterministic sampling. We note that our framework uses DDIM with $\eta=0$ (deterministic sampling) during inference, meaning the model can produce a single deterministic output when needed. The diffusion framework thus provides flexibility: it can generate deterministic predictions for practical use while being trained on the full distribution of neural responses.
>
> **W2: Justification for DiT over UNet-based diffusion in neuroscience.**
>
> We agree that the superiority of DiT over UNet has not been previously validated in neuroscience. We have added a UNet-based diffusion model baseline in our response to Reviewer NPdM's second question for direct comparison. Beyond the empirical comparison, we provide the following architectural motivation:
>
> - In our framework, M/EEG signals are patchified into spatio-temporal tokens with brain region embeddings and temporal embeddings. The Transformer's global self-attention allows every patch to attend to all other patches, which is well-suited for modeling spatio-temporal dependencies across brain regions and time.
> - In contrast, UNet relies on local convolutions with limited receptive fields at each layer. While effective for spatially structured images, the 2D structure of brain signals (channels × time) does not have the same spatial locality properties as natural images.
> - Furthermore, the cross-attention mechanism in our DiT blocks provides a natural interface for conditioning on the CLIP embeddings, which is more architecturally elegant than the typical conditioning approaches in UNet-based diffusion.
>
> **W3: Novelty concern — "plug-and-play" approach.**
>
> The contributions of our work are as follows:
>
> 1. Problem-level contribution: We are among the first to formalize image-to-M/EEG encoding as a conditional generative task and demonstrate that diffusion models can produce biologically plausible brain signals. This problem formulation itself is a contribution to the field.
>
> 2. Cross-attention design for cross-modal alignment: The design of using brain signal patches as Query and unified visual-semantic embeddings as Key/Value is a principled choice — it allows the image information to modulate the brain signal generation process, mirroring how visual stimuli modulate neural activity.
>
> 3. Spatio-temporal position encoding: We propose the spatio-temporal position encoding based on the bottom-up hierarchy of visual processing in neuroscience. The brain region ablation study demonstrates that these embeddings capture meaningful neuroscience structure.
>
> 4. LLM-enhanced semantic conditioning: The integration of LLM-generated captions provides semantic information that pure visual features lack. The ablation study confirms the benefit of this component.
>
>
> **W4: Cross-subject generalization and subject-specific adapters.**
>
> We fully agree with this insightful suggestion. The topics you mentioned, are current research hotspots and promising directions for us to explore in depth. Due to the limited rebuttal period, it is difficult for us to develop effective subject-specific adapters or neural alignment layers in a short time. Instead, we have adopted a leave-one-subject-out (LOSO) training strategy to increase the training data, allowing us to demonstrate the within-subject and cross-subject performance of our framework with a larger data volume. The experimental results can be found in our response to Reviewer NPdM's second question.

---

> > ### Author Rebuttal · Reviewer_HfVL · 2026-04-03
> >
> > Thank the authors for the detailed clarification and additional discussion. The response helps contextualize the motivation for using a diffusion-based framework. However, a key concern remains regarding whether the modeled stochasticity is functionally meaningful for neural encoding.
> >
> > The authors justify the use of a Diffusion framework by citing the "one-to-many" stochastic nature of EEG/MEG data. However, this argument fails to distinguish between stochastic measurement and stochastic encoding:
> > * In neuroscience, the "stochasticity" observed across 80 trials of the same image (THINGS-EEG2) is primarily attributed to non-stimulus-related factors, such as thermal noise, muscle artifacts, etc. Encoding models are traditionally designed and expected to isolate the evoked response $R_\\text{evoked}$ from this background noise. By intentionally modeling the "full distribution", the authors are capturing variance that is, by definition, not stimulus-dependent.
> > * The claim that a deterministic model "loses information" by predicting the conditional mean ignores the fact that $E[R|S]$ is the most reliable estimate of the stimulus-locked neural representation. Unless the authors can demonstrate that the variance captured by their Diffusion model is stimulus-contingent (i.e., representing neural uncertainty rather than instrumental noise), the model lacks functional relevance.
> > * A generative model that produces "diverse" neural responses is only neuroscientifically valid if that diversity is trial-aligned. If the model's samples do not correlate with the specific trial-level fluctuations of the subject, the model is simply a sophisticated noise-regressor.
> > * If the authors' final recommendation is to use DDIM for deterministic inference ($\eta=0$), they have effectively reduced their generative model to a standard regressor. The authors have to show why this complex generative overhead is superior to a high-capacity Transformer that targets the evoked signal directly.
> >
> > The current justification for the diffusion framework would be significantly strengthened by empirical evidence demonstrating that the modeled variability is stimulus-dependent and that capturing the full response distribution provides measurable benefits beyond standard deterministic or probabilistic regression approaches.

---

> > > ### Author Response · Authors · 2026-04-03
> > >
> > > We thank the reviewer for the continued discussion and the opportunity to strengthen our justification. Building on our previous response, we now provide additional empirical evidence addressing both concerns raised by the reviewer.
> > >
> > > **Evidence 1: The modeled variability is stimulus-dependent.**
> > >
> > > In addition to the arguments in our previous response, we have now conducted a mismatch control experiment (also provided to Reviewer jCbw). Both THINGS-EEG2 and THINGS-MEG test sets contain 200 unique test images. We compute the Pearson correlation between generated signals and real brain signals under two conditions: (1) *matched* — the generated and real signals correspond to the same image; (2) *mismatched* — the generated signal for image $i$ is paired with the real brain signal of a randomly selected different image $j$ ($i \neq j$), repeated 10 times for stability.
> > >
> > > | Setting | EEG Pearson | MEG Pearson |
> > > |---------|-------------|-------------|
> > > | Matched (original) | 0.425 | 0.379 |
> > > | Mismatched (control) | 0.027 | 0.012 |
> > >
> > > The mismatched control yields near-zero Pearson correlations (EEG: 0.027, MEG: 0.012), while the matched scores are substantially higher (EEG: 0.425, MEG: 0.379).
> > > This large gap (0.398 for EEG, 0.367 for MEG) provides direct empirical evidence that our generated signals are stimulus-dependent.
> > >
> > > **Evidence 2: Benefits of the diffusion framework beyond standard regression.**
> > >
> > > To directly address the reviewer's question, we design a controlled ablation: **Direct Regression Baseline (w/o Diffusion)**. This baseline uses the *exact same* architecture as our method — the same DiT blocks, cross-attention mechanism (with CLIP image and text embeddings as Key/Value), spatio-temporal position encoding, and brain region embeddings — but removes the diffusion process entirely. Instead of learning to denoise across multiple noise levels, the model directly predicts the brain signal from CLIP embeddings in a single forward pass, trained with a standard MSE regression loss. All other hyperparameters (learning rate, optimizer, batch size, training epochs) are kept identical. This isolates the effect of the diffusion training paradigm as the *only* variable.
> > >
> > > *Table 1: Diffusion vs. Direct Regression on THINGS-EEG2*
> > >
> > > | Method | MSE ↓ | Pearson ↑ | Cosine ↑ | SL ↑ |
> > > |--------|-------|-----------|----------|------|
> > > | Direct Regression (w/o Diffusion) | 0.168 | 0.331 | 0.324 | 0.531 |
> > > | **Our work (Diffusion)** | **0.109** | **0.425** | **0.402** | **0.614** |
> > >
> > > *Table 2: Diffusion vs. Direct Regression on THINGS-MEG*
> > >
> > > | Method | MSE ↓ | Pearson ↑ | Cosine ↑ | SL ↑ |
> > > |--------|-------|-----------|----------|------|
> > > | Direct Regression (w/o Diffusion) | 0.771 | 0.261 | 0.258 | 0.456 |
> > > | **Our work (Diffusion)** | **0.663** | **0.379** | **0.382** | **0.597** |
> > >
> > >
> > > Since the only difference between the two models is the training paradigm (diffusion denoising vs. direct MSE regression), any performance gap can be directly attributed to the diffusion framework itself.
> > > The results confirm that the diffusion training paradigm provides measurable benefits beyond standard regression.

---

### Official Review · Reviewer_jCbw · 2026-03-13

**Soundness:** 3
**Presentation:** 3
**Significance:** 2
**Originality:** 2
**Overall Recommendation:** 4
**Confidence:** 3

**Summary:**

The paper proposes an image-to-brain signal generation framework designed for the encoding stage of visual prostheses. The authors utilize a Diffusion Transformer (DiT) architecture based on Denoising Diffusion Implicit Models (DDIM) to convert images into biologically plausible M/EEG signals. To capture both visual and semantic information, the model aligns brain signal embeddings with unified visual-semantic embeddings (concatenated CLIP image and text embeddings, the latter derived from LLM-generated captions) using cross-attention. Additionally, they introduce a learnable spatio-temporal position encoding to model the inherent structure of brain data. Experiments on the THINGS-EEG2 and THINGS-MEG datasets show the framework outperforms traditional and generative baselines

**Compliance With Llm Reviewing Policy:**

Affirmed.

**Ethical Review Concerns:**

No Ethical Review Concerns

**Final Justification:**

Authors have provided enough evidence to address most of my concerns throughout the rebuttal. The level of impact of the paper does not justify increasing my score to 5 but I will maintain my initially high score of 4.

**Key Questions For Authors:**

- How do the authors explicitly ensure there is no information leakage between the brain signal patches used as Queries and the target reconstructed signal?
- Given the relatively low spatial resolution of EEG and MEG, how do the authors justify the precision of your "brain region" embeddings, and how are these regions mapped for different subjects?
- Why was SynBrain the only recent generative baseline utilized? Are there other diffusion-based encoding models (e.g., those used in fMRI) that could have been adapted for a more robust comparison?
- Can the authors review more extensively the litterature in the field

**Limitations:**

The authors discuss inter-individual variability and the challenges of cross-subject generalization. However, they should more explicitly address the ethical implications of using generative models to "predict" neural states in blind individuals.

**Strengths And Weaknesses:**

Soundness
+ The integration of LLMs for generation enhancement ( + use of DDIM for faster sampling) are wise choices that address common bottlenecks in generative modeling for neural data.
- The claim of "biological plausibility" is primarily supported by metric improvements (MSE, Pearson) and a brain region ablation study. However, the paper lacks a clear explanation of how the model ensures no data leakage between the input neural data and reconstructed signals during training.

Presentation
+ The overall architecture (Figure 2) and the problem definition are clearly presented.
- The manuscript contains several unprofessional formatting issues, such as retaining the "Submission and Formatting Instructions" header on multiple pages. Some descriptions of the visual prosthesis pipeline are circular or confusingly phrased (first paragraphs of introduction).

Originality:
+ Combining DiT with LLM-generated captions for brain signal alignment is a creative application of state-of-the-art generative techniques to neuroscience.
- The hierarchical processing argument (V1 to IT) is a well-established concept in the field, and the bibliography regarding this is incomplete. For example, “deep neural network features have been shown to predict neural responses across the ventral visual stream, with early layers corresponding to V1 and deeper layers to inferotemporal cortex”bibliography incomplete, cite among others "Disentangling the factors of convergence between brains and computer vision models"

Significance
+ Addressing the "brain encoding" stage is a crucial and under-explored step toward a complete visual prosthesis pipeline.
- Some results need to be proved/made explicit:
the "multimodal" claim is slightly overstated: 1) applying the same architecture to two separate modalities (EEG and MEG) is different from a truly multimodal data integration approach. 2) “brain region” is overstatement for MEG and EEG? does not seem spatially accurate enough.
Can you make more explicit how you ensure no leakage between the neural data given as input to the model and the reconstructed neural signal
“Furthermore, our method also outperforms the recent generative approach SynBrain” are you confident this is the only recent baseline you can compare yourselves to? if not, the paper needs adding of more recent baselines

---

> ### Author Rebuttal · Authors · 2026-03-30
>
> We thank the reviewer for the constructive feedback. We address each question below.
>
> **Q1: No information leakage between Query and target signal.**
>
> Our framework follows the standard diffusion model paradigm, and there is no information leakage. Specifically:
>
> - During training, the Query is derived from the noisy brain signal $\mathbf{y}_t = \sqrt{\bar{\alpha}_t}\mathbf{y}_0 + \sqrt{1-\bar{\alpha}_t}\boldsymbol{\epsilon}$, where the noise level is controlled by the randomly sampled timestep $t$. The model is trained to predict the added noise $\boldsymbol{\epsilon}$, not to reconstruct $\mathbf{y}_0$ directly. The noisy input $\mathbf{y}_t$ at large $t$ is dominated by Gaussian noise with minimal information from $\mathbf{y}_0$.
>
> - During inference, the Query is initialized from pure Gaussian noise $\mathbf{y}_T \sim \mathcal{N}(\mathbf{0}, \mathbf{I})$, which contains zero information from any real brain signal. The model iteratively denoises this noise, conditioned solely on the CLIP image and text embeddings (Key and Value in cross-attention). Therefore, the generated brain signal is entirely driven by the visual input, with no access to any ground-truth brain data.
>
> - The conditioning information (Key and Value) comes exclusively from the input image through CLIP encoders and LLM-generated captions — these are completely independent of the brain signal data.
>
> **Q2: Justification of "brain region" embeddings given low spatial resolution.**
>
> Our "brain region" embeddings operate at a macroscopic lobe-level partition, not fine-grained cortical localization:
>
> - For EEG (63 channels), we group electrodes into 5 coarse regions (Frontal, Central, Temporal, Parietal, Occipital) based on the standard international 10-20 system electrode naming convention. This grouping is a well-established practice in the EEG literature (e.g., Frontal: Fp1, Fp2, F3, F4...; Occipital: O1, Oz, O2). The 10-20 system provides a standardized mapping that is consistent across subjects since all subjects wear the same standard electrode cap.
>
> - For MEG (271 channels), we group magnetometer sensors into the same 5 macroscopic regions based on their physical positions on the MEG helmet. Since all participants use the same MEG device (CTF 275-channel system), the sensor-to-region mapping is also consistent across subjects.
>
> - Our brain region ablation study (Figure 6) provides empirical validation: removing the occipital region causes the largest performance drop, which aligns with the well-established neuroscience knowledge that the occipital cortex is the primary visual processing area. This would not be observed if the region embeddings were arbitrary.
>
> We clarify that "multimodal" in our paper refers to the fusion of images, text, and brain signals, not EEG and MEG fusion.
>
> **Q3: Why SynBrain was the only recent generative baseline.**
>
> Please refer to our consolidated response to Reviewer NPdM's Q2.
>
> **Q4: More extensive literature review.**
>
> We thank the reviewer for this suggestion.
>
> We will add the suggested reference "Disentangling the factors of convergence between brains and computer vision models" to the position mentioned in the weakness. Furthermore, "Toward a realistic model of speech processing in the brain with self-supervised learning" and "Brains and algorithms partially converge in natural language processing" also demonstrate that the internal representations of deep learning models are related to those of the human brain. We will also add these two references.
>
> Furthermore, we discuss recent advances in brain encoding models. "Disentangling Superpositions: Interpretable Brain Encoding Model with Sparse Concept Atoms" addresses the feature entanglement problem in encoding models by transforming dense embeddings into sparse concept atoms, offering an interpretable bridge between artificial neural network features and brain representations. "MindSimulator: Exploring Brain Concept Localization via Synthetic FMRI" leverages generative models to synthesize fMRI conditioned on visual stimuli, sharing our motivation of generative brain signal synthesis.
>
> Due to the word limit, we are unable to include all relevant works. We sincerely appreciate the reviewer's valuable suggestion.
>
> **Presentation weakness**
>
> We have revised the presentation issues in the manuscript. As for the concern about the first paragraphs of the introduction being "circular or confusingly phrased," we would like to ask whether this refers to the phrase "evoking visual perception (a pattern of localized light flashes, 'visual percept', or 'phosphene')." If so, our intention is to introduce multiple expressions for the same concept for interchangeable use in subsequent text. If the reviewer finds this inappropriate, we will revise the phrasing accordingly.

---

> > ### Author Rebuttal · Reviewer_jCbw · 2026-04-03
> >
> > Thank you for this rebuttal and additional experiments with two new baselines. Could you please provide an explanation as to why LOSO (Cross-subject) sometimes gives better performance (Cosyne) than without LOSO? The reviewer is also curious to see future results on THINGS-EEG2.
> >
> > The reviewer is unsure of the relevance of the papers "Toward a realistic model of speech processing in the brain with self-supervised learning" and "Brains and algorithms partially converge in natural language processing" referring to language models, to support the sentence “deep neural network features have been shown to predict neural responses across the ventral visual stream, with early layers corresponding to V1 and deeper layers to inferotemporal cortex” in a paper about vision, can the authors please explicit their reasoning?
> >
> > Regarding the compelling argument of reviewer HfVL and your response, “If the generated signals were not stimulus-dependent, these metrics would be near zero”, could you please provide a control for this and the resulting pearson scores for MEG and EEG, that would serve as baseline for your original pearson scores ?

---

> > > ### Author Response · Authors · 2026-04-06
> > >
> > > **AQ1: Why LOSO (Cross-subject) sometimes gives better performance than without LOSO, and THINGS-EEG2 results.**
> > >
> > > Due to the 5000-character limit of this response box, we are unable to include the full explanation here. We have provided the THINGS-EEG2 LOSO results and a detailed explanation based on topographic analysis in our response to Reviewer NPdM's acknowledgement below. We kindly ask the reviewer to refer to that section for the complete answer.
> > >
> > > **AQ2: Relevance of the two cited language/speech papers.**
> > >
> > > We thank the reviewer for pointing this out. Upon further reflection, we acknowledge that citing “Toward a realistic model of speech processing in the brain with self-supervised learning” and “Brains and algorithms partially converge in natural language processing” in the context of *visual* neural encoding is not sufficiently precise. Our original intention was to highlight the broader principle that deep neural network representations — across multiple domains — exhibit structural correspondence with brain representations, thereby motivating the use of DNN features as conditioning signals for brain signal generation. However, we agree with the reviewer that these two papers focus specifically on the language and speech domains, and their findings do not directly support the specific claim about the *ventral visual stream hierarchy* (V1 to inferotemporal cortex).
> > >
> > > We appreciate the reviewer’s careful scrutiny. Given that these two references are not directly relevant to the visual domain claim, we will withdraw our plan to include them in the manuscript to maintain precise alignment between citations and claims.
> > >
> > > **AQ3: Control experiment for stimulus-dependence of generated signals.**
> > >
> > > We thank the reviewer for this constructive suggestion. We agree that providing a control baseline is important for substantiating the stimulus-dependence of our generated signals. We have now conducted the requested control experiment.
> > >
> > > **Control setup:** Both THINGS-EEG2 and THINGS-MEG test sets contain 200 unique test images. For each test image, we compute the Pearson correlation between its generated signal and the real brain signal (averaged over 80 repetitions for EEG, 12 repetitions for MEG) — this is the *matched* score. For the *mismatched* control, we pair the generated signal for test image $i$ with the real brain signal of a randomly selected different test image $j$ ($i \neq j$), and repeat this random mismatching 10 times to obtain a stable average. If the generated signals were not stimulus-dependent, the matched and mismatched Pearson scores should be comparable; conversely, a large gap directly quantifies the stimulus-dependent component.
> > >
> > > **Results:**
> > >
> > > | Setting | EEG Pearson | MEG Pearson |
> > > |---------|-------------|-------------|
> > > | Matched (original) | 0.425 | 0.379 |
> > > | Mismatched (control) | 0.027 | 0.012 |
> > >
> > > The mismatched control yields near-zero Pearson correlations for both EEG (0.027) and MEG (0.012), confirming that the high Pearson scores in our original evaluation (0.425 and 0.379) are indeed driven by stimulus-specific information rather than any general structural similarity between generated and real signals. The large gap between matched and mismatched scores (0.398 for EEG, 0.367 for MEG) provides strong quantitative evidence that our generated signals are stimulus-dependent.

---

### Decision · Program_Chairs · 2026-04-30

**Decision:**

Accept (regular)

**Comment:**

The paper tackles an important and underexplored problem of image-to-brain signal generation for visual prosthesis, and proposes a diffusion-based framework with multimodal conditioning that achieves strong empirical performance. Reviewers appreciated the novelty of addressing the brain encoding stage, the clear technical formulation, and consistent gains across datasets, though concerns were raised about the use of diffusion models, limited baseline comparisons, and cross-subject generalization. The authors provided detailed rebuttals with additional experiments, stronger baselines, and clarifications, which resolved most concerns for the majority of reviewers. Despite one review of score 2, the authors have followed up. The concensus is generally positive.